# ENABLING LANGUAGE MODELS TO IMPLICITLY LEARN SELF-IMPROVEMENT

**Ziqi Wang**[1][*] **Le Hou**[2][†] **Tianjian Lu**[2]**, Yuexin Wu**[2]**, Yunxuan Li**[2]**, Hongkun Yu**[2]**, Heng Ji**[1]
[1] University of Illinois Urbana-Champaign [2] Google
`ziqiw9@illinois.edu lehou@google.com`

## ABSTRACT

Large Language Models (LLMs) have demonstrated remarkable capabilities in open-ended text generation tasks. However, the inherent open-ended nature of these tasks implies that there is always room for improvement in the quality of model responses. To address this challenge, various approaches have been proposed to enhance the performance of LLMs. There has been a growing focus on enabling LLMs to self-improve their response quality, thereby reducing the reliance on extensive human annotation efforts for collecting diverse and high-quality training data. Recently, prompting-based methods have been widely explored among self-improvement methods owing to their effectiveness, efficiency, and convenience. However, those methods usually require explicitly and thoroughly written rubrics as inputs to LLMs. It is expensive and challenging to manually derive and provide all necessary rubrics with a real-world complex goal for improvement (e.g., being more helpful and less harmful). To this end, we propose an Im**P**licit Self-**I**mprovemen**T** (PIT) framework that implicitly learns the improvement goal from human preference data. PIT only requires preference data that are used to train reward models without extra human efforts. Specifically, we reformulate the training objective of reinforcement learning from human feedback (RLHF) – instead of maximizing response quality for a given input, we maximize the quality gap of the response conditioned on a reference response. In this way, PIT is implicitly trained with the improvement goal of better aligning with human preferences. Experiments on two real-world datasets and one synthetic dataset show that our method significantly outperforms prompting-based methods.

## 1 INTRODUCTION

LLMs (Devlin et al., 2018; Raffel et al., 2020; Brown et al., 2020; Chowdhery et al., 2022; Schulman et al., 2022; OpenAI, 2023; Anil et al., 2023) have achieved state-of-the-art results on complex tasks such as math reasoning (Wei et al., 2022; Xue et al., 2023; Zhou et al., 2023), summarization (Stiennon et al., 2020b), conversations (Schulman et al., 2022; Bai et al., 2022), schema induction (Li et al., 2023) and solving domain-specific problems (Singhal et al., 2022). The keys of LLMs success are their abilities of following instructions and aligning with human preferences (Ouyang et al., 2022; Peng et al., 2023a; Shen et al., 2023). A widely adopted approach toward them is instruction fine-tuning and reinforcement learning from human feedback (RLHF) (Ouyang et al., 2022). However, instruction fine-tuning and RLHF are imperfect, and there is always room for improvement. For example, LLMs may hallucinate information (OpenAI, 2023), have reasoning errors (Bubeck et al., 2023), and generate unhelpful and harmful contents (Bai et al., 2022). A straightforward approach is to collect more diverse and high-quality data and improve the alignment with a human-in-the-loop training paradigm (Ouyang et al., 2022), which requires extensive amount of human effort, especially for specific domains that require expert knowledge.

Therefore, the community has explored to use LLMs to self-improve their own response quality without human intervention. With the advent of generative language models, prompting methods have proved effective and efficient (no need to train models) with convenience (no need to serve models and can be used with black-box language models through APIs). Madaan et al. (2023) use

---

[*]Work done when interning at Google
[†]Correspondence Author

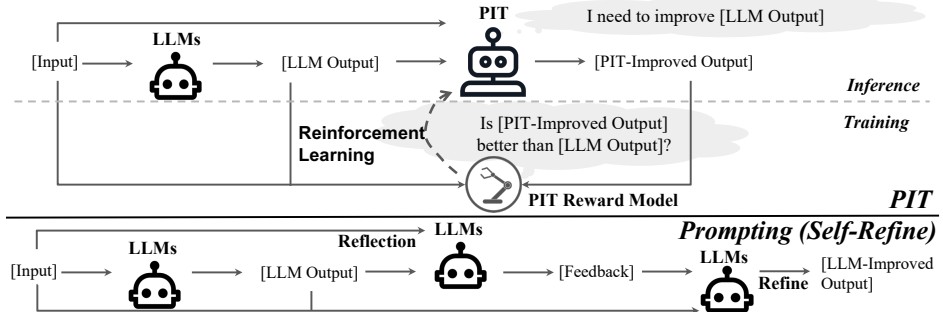

Figure 1: The pipeline of PIT and prompting methods (Self-Refine). **Upper**: PIT utilizes inputs and LLM outputs for further improvement during inference. During training, a reward model will assess the gap of PIT-improved outputs and LLM outputs, and the reward is used for reinforcement learning to train PIT. **Lower**: Self-Refine uses LLMs to give self-feedback and then asks LLMs to improve outputs based on the self-feedback.

LLMs to give self-feedback and iteratively improve LLMs response quality, Chen et al. (2023) enable LLMs to self-debug to enhance code generations. Nevertheless, self-improvement in language models through prompting can be less than ideal, as humans often find it challenging to define comprehensive improvement goals and create detailed assessment rubrics. For example, improving helpfulness and harmlessness of LLMs requires careful definitions of these qualities to guide LLMs in improving their responses accordingly. As a result, prompting could make LLMs self-improve well only if the improvement goal is clear, simple, and well-defined through prompting. LLMs perform poorly in our experiments if we ask them to make responses `more helpful`. This is because the LLM tends to add more details to the response and thus makes the response longer, whereas more details are not always preferred since it may lead to the response going off the topic. Instead, asking language models to `be polite, offer necessary information, and avoid going off topics` will yield better improvements. Previous methods have also discussed similar observations on the effect of detailed rubrics. Rafailov et al. (2023) showed that using a simple rubric of `Which summary is better?` will yield much more disagreement with humans than a more comprehensive rubric of `Which summary summarizes most important points without including unimportant details?`

Despite the effectiveness, detailed rubrics are often hard to obtain. First, it is hard for humans to infer and write all possible rubrics. Second, when the task requires domain expertise (e.g., clinical domain (Singhal et al., 2022)), it is impractical to scale up. To this end, we switch our focus from explicitly designing rubrics to implicitly learning self-improvement from data. We notice that the preference data used to train reward models implicitly tells how to improve the response quality. Thus, we utilize this preference data to train a novel model that implicitly understands self-improvement goals from data. Using this method, we eliminate the need for rubric design and avoid the need for additional data since we only reuse the data used to train reward models.

We denote our approach Im**P**licit Self-**I**mprovemen**T** (PIT), a novel approach that enables the model to learn self-improvement implicitly from data. Specifically, we reformulate the instruction fine-tuning and RLHF process and switch the training objective from maximizing response quality for given inputs to maximizing response quality gap conditioned on reference responses for given inputs. Figure 1 shows the working flows of PIT and prompting methods. PIT utlizes given inputs and reference responses and generates improved responses accordingly. Similar to prompting methods, PIT can repeat this process iteratively by replacing reference responses with improved responses. Compared with prompting methods, our method PIT does not require a manual rubric design. Extensive evaluations on two real-world datasets and one synthetic dataset show the effectiveness of PIT compared with prompting methods such as Self-Refine (Madaan et al., 2023).

## 2 RELATED WORK

**Alignment** Alignment is critical for a helpful and harmless language model (Ouyang et al., 2022; Bai et al., 2022). One common way for alignment is RLHF (Ouyang et al., 2022). However, RLHF is

sensitive to the training details and is complicated to implement. To get rid of the RL, Lu et al. (2022) use quantized rewards as control tokens to continually fine-tune policy models; Diao et al. (2023); Dong et al. (2023); Gulcehre et al. (2023) use reward models to filter out high-reward generations for further fine-tuning . Some approaches are even simpler and do not require reward models. Liu et al. (2023) use human feedback as control tokens directly to fine-tune models on model generations. Sun et al. (2023) use human-written principles to guide language models to generate helpful and harmless responses and use generated responses to fine-tune models. Zhao et al. (2023) use a calibration loss to mimic RLHF. Rafailov et al. (2023) designs a loss function that can be theoretically proved to be identical to RLHF. Xiong et al. (2023) shows that online alignment is much better than offline alignment. We used RLHF for alignment in our paper, as it is the most commonly used method.

**Self-improvement** Self-improvement enables language models to improve themselves without extra human effort. Moreover, the self-improved responses can then be utilized for context distillation (Askell et al., 2021) to update LLMs. Huang et al. (2022) use PaLM (Chowdhery et al., 2022) to label more task-specific data and use the most confident labeled data to continue fine-tuning PaLM itself. However, fine-tuning language models on specific tasks may lower the overall generation performance of models (Zhai et al., 2023). Therefore, researchers also put much effort into self-improvement without modifying language models themselves. Shinn et al. (2023) put language models into an environment and let language models reflect their actions when they meet failures. Wang et al. (2023a) use the Python interpreter's error message to improve language models' code generation abilities. However, these methods require environments that can provide automatic feedback, which may not be accessible in general. Self-Refine (Madaan et al., 2023) enables LLMs to reflect on their responses and provide feedback, then ask LLMs to use feedback to improve responses. Zeng et al. (2023) apply Self-Refine to generate better meta-reviews for papers. Xue et al. (2023) guide LLMs to fine-grained feedback for math problems. The feedback could also benefit from tools. Zhou et al. (2023) use OpenAI's code interpreter to collect more accurate feedback. Wang et al. (2023b) propose a benchmark to evaluate how models can get improved from human or AI feedback. Nevertheless, these methods require explicit prompting to collect feedback and self-improve. Our method does not require modifying model weights, interactive environment, or explicitly prompting.

## 3 METHOD

Policy models (i.e., "LLMs" in Figure 1) trained with RLHF generate reference responses for given inputs, whereas PIT takes the given inputs and reference responses as its inputs and generates improved responses. The different input formats between policy models and PIT (See Appendix A for details) requires reformulating RLHF training objectives to train PIT. We follow and reformulate RLHF steps proposed by Ouyang et al. (2022): supervised fine-tuning, reward model training, and reinforcement learning. Although there are many other alignment methods such as Direct Preference Optimization (Rafailov et al., 2023) or Preference Ranking Optimization (Song et al., 2023), we choose RLHF to train policy models and PIT for two reasons: (1) RLHF is the most widely used approach for alignment (2) PIT is not aimed for alignment but for triggering self-improvement. Therefore, the alignment method is not the focus of this paper.

### 3.1 FORMULATION

Suppose we have data $\mathcal{D} = \{(x, y_l, y_w)\}_{3n}$, where $x$ is the input prompt, $y_l$ is the worse model generation, and $y_w$ is the better model generation, annotated by humans. We could equally divide the data into three folds $\mathcal{D}_{\text{SFT}}$, $\mathcal{D}_{\text{RM}}$ and $\mathcal{D}_{\text{RL}}$ for the supervised fine-tuning, reward model training, and reinforcement learning, respectively. The policy model $M_P$ is trained to generate response $y_P$ for the given input $x$: $y_P \sim M_P(\cdot|x)$. PIT model $M_{\text{PIT}}$ is trained to generate improved response $y_{\text{PIT}}$ for given input $x$ and reference response $y_{\text{ref}}$: $y_{\text{PIT}} \sim M_{\text{PIT}}(\cdot|x, y_{\text{ref}})$. In the following sections, we use subscript $\cdot_P$ to denote the policy model and $\cdot_{\text{PIT}}$ to denote PIT, and use superscripts such as SFT or RL to denote different checkpoints of models.

### 3.2 SUPERVISED FINE-TUNING

Supervised fine-tuning (SFT) is the pre-requisite step of RLHF (Ouyang et al., 2022). SFT only uses $x$ and $y_w$, and $y_w$ could also be human-written responses Ouyang et al. (2022). To train policy models, we follow Ouyang et al. (2022) and simply maximize the likelihood of $y_w$: $\mathcal{L}_P^{\text{SFT}} = -\sum_{(x,y_l,y_w)\in\mathcal{D}_{\text{SFT}}} \log M_P(y_w|x)$.

Since PIT aims to improve reference responses, we need to include $y_l$ to train PIT: $\mathcal{L}_{\text{PIT}}^{\text{SFT}} = -\sum_{(x,y_l,y_w)\in\mathcal{D}_{\text{SFT}}} \log \text{M}_{\text{PIT}}(y_w|x,y_l)$. A natural extension for this loss function is that we can apply unlikelihood (Welleck et al., 2019) loss to $\text{M}_{\text{PIT}}(y_l|x,y_w)$, $\text{M}_{\text{PIT}}(y_l|x,y_l)$ and $\text{M}_{\text{PIT}}(y_w|x,y_w)$. However, we find the unlikelihood loss causes performance degradation in experiments, therefore we only keep likelihood loss when training PIT with SFT.

## 3.3 REWARD MODEL TRAINING

The reward model is used to judge how good a response is and is the key component for reinforcement learning. A reward model denoted as $\text{R}_{\text{P}}$ maps an input $x$ and the corresponding response $y$ to a scalar $r$: $r(x,y) = \text{R}_{\text{P}}(x,y)$. Since the data $\mathcal{D}_{\text{RM}}$ only contains preference ($y_l$ and $y_w$) and does not provide $r$ directly, Ouyang et al. (2022) trains the reward model by maximizing the reward gap between $y_l$ and $y_w$: $\mathcal{L}_{\text{P}}^{\text{RM}} = -\sum_{\mathcal{D}_{\text{RM}}} \log \sigma(r_w - r_l)$, where $r_w$ and $r_l$ are rewards, i.e., $\text{R}_{\text{P}}(x,y_w)$ and $\text{R}_{\text{P}}(x,y_l)$, respectively. Unlike $\text{R}_{\text{P}}$, the reward model $\text{R}_{\text{PIT}}$ does not focus on rewards $r$ of responses but cares about the gap $r_{\text{gap}}$ between responses: $r_{\text{gap}}(x,y_1,y_2) = \text{R}_{\text{PIT}}(x,y_1,y_2)$. Therefore, $\text{R}_{\text{PIT}}$ needs to learn $(r_w - r_l)$ from the training data $\mathcal{D}_{\text{RM}}$. To formulate the loss function, we can use the fact that

$$r_{\text{gap}}(x,y_w,y_l) \geq r_{\text{gap}}(x,y_w,y_w) \approx r_{\text{gap}}(x,y_l,y_l) \geq r_{\text{gap}}(x,y_l,y_w). \tag{1}$$

To ensure that the reward model performs as expected in Equation 1, after training, we consider all pairwise relations mentioned above, and the loss function becomes:

$$\begin{aligned} \mathcal{L}_{\text{PIT}}^{\text{RM}} = -\sum_{\mathcal{D}_{\text{RM}}} [&\log \sigma(r_{\text{gap}}^{w,l} - r_{\text{gap}}^{w,w}) + \log \sigma(r_{\text{gap}}^{w,l} - r_{\text{gap}}^{l,l}) + \log \sigma(r_{\text{gap}}^{w,l} - r_{\text{gap}}^{l,w}) \\ &+ \log \sigma(r_{\text{gap}}^{w,w} - r_{\text{gap}}^{l,w}) + \log \sigma(r_{\text{gap}}^{l,l} - r_{\text{gap}}^{l,w})], \end{aligned} \tag{2}$$

where $r_{\text{gap}}^{w,l}$ is the shortcut of $r_{\text{gap}}(x,y_w,y_l)$, etc. Although there are other options to model the reward gap, such as computing the rewards subtraction through $\text{R}_{\text{P}}$, we find Equation 2 is the best fit. More discussions could be found in Appendix C.

## 3.4 REINFORCEMENT LEARNING

Reinforcement learning (RL) finds a policy that maximizes expected rewards over time, aligning LLMs with human preferences. The optimization goal for RL is (Ouyang et al., 2022):

$$\text{Optimization}_{\text{P}}^{\text{RL}} = \sum_{\mathcal{D}_{\text{RL}}} \left[ r(x,y) - \beta \text{KL}(\text{M}_{\text{P}}^{\text{RL}}(y|x) - \text{M}_{\text{P}}^{\text{SFT}}(y|x)) \right]. \tag{3}$$

where $r(x,y) = \text{R}_{\text{P}}(x,y)$ and $y \sim \text{M}_{\text{P}}^{\text{RL}}(\cdot|x)$. $\text{M}_{\text{P}}^{\text{RL}}$ is the policy model to be optimized, which is initialized to $\text{M}_{\text{P}}^{\text{SFT}}$. $\text{M}_{\text{P}}^{\text{SFT}}$ is $\text{M}_{\text{P}}$ trained in supervised fine-tuning, which is fixed in the RL. The KL divergence is used to restrict dramatic weight changes and reward hacking in RL.

### 3.4.1 CURRICULUM REINFORCEMENT LEARNING

Different from Equation 3, PIT aims to improve a reference response $y_{\text{ref}}$ instead of generating a response from scratch. Therefore, the RL for PIT utilizes $x$ and $y_{\text{ref}}$ simultaneously. The difficulty is how to choose $y_{\text{ref}}$. An intuitive way is to use $y_l$ and $y_w$ provided in the dataset, and the optimization goal becomes:

$$\text{Optimization}_{\text{PIT}}^{\text{RL, 0}} = \sum_{\mathcal{D}_{\text{RL}}} \sum_{y_{\text{ref}}\in\{y_l,y_w\}} \left[ r_{\text{gap}}(x,y,y_{\text{ref}}) - \beta \text{KL}(\text{M}_{\text{PIT}}^{\text{RL}}(y|x,y_{\text{ref}}) - \text{M}_{\text{PIT}}^{\text{SFT}}(y|x,y_{\text{ref}})) \right]. \tag{4}$$

However, PIT aims to improve $\text{M}_{\text{P}}^{\text{RL}}$ responses, and $y_l$ and $y_w$ are chosen from the annotated data, not sampled from the $\text{M}_{\text{P}}^{\text{RL}}$. Therefore, we need to do another round of reinforcement learning, where $y_{\text{ref}}$ is sampled from the policy model:

$$\text{Optimization}_{\text{PIT}}^{\text{RL, 1}} = \sum_{\mathcal{D}_{\text{RL}}} \sum_{y_{\text{ref}}\sim\text{M}_{\text{P}}^{\text{RL}}(\cdot|x)} \left[ r_{\text{gap}}(x,y,y_{\text{ref}}) - \beta \text{KL}(\text{M}_{\text{PIT}}^{\text{RL}}(y|x,y_{\text{ref}}) - \text{M}_{\text{PIT}}^{\text{SFT}}(y|x,y_{\text{ref}})) \right]. \tag{5}$$

The flexibility of $y_{\text{ref}}$ in fact enables us to do multiple rounds of reinforcement learning to improve PIT further. For example, a third round $\text{Optimization}_{\text{PIT}}^{\text{RL, 2}}$ can sample $y_{\text{ref}} \sim \text{M}_{\text{PIT}}^{\text{RL}}(\cdot|x,y_1)$, where $y_1 \sim \text{M}_{\text{P}}^{\text{RL}}(\cdot|x)$ with other terms unchanged. The third round enables PIT to improve the improved $\text{M}_{\text{P}}^{\text{RL}}$ responses. In principle, this process could be extended to infinite rounds. We denote this process as curriculum reinforcement learning.

### 3.4.2 DISCUSSIONS ON CURRICULUM REINFORCEMENT LEARNING

It is worth noting that the first round $\text{Optimization}_{\text{PIT}}^{\text{RL, 0}}$ is necessary since the optimization $\text{Optimization}_{\text{PIT}}^{\text{RL, 1}}$ is too hard to be optimized directly. This is because responses sampled from $\text{M}_{\text{P}}^{\text{RL}}$ are already of high quality (higher than $y_l$ and $y_w$ in the data, as shown in Section 4.6), and are harder to be improved compared to $y_l$ and $y_w$. Therefore, curriculum reinforcement learning is needed to ensure the success of optimizing $\text{Optimization}_{\text{PIT}}^{\text{RL, 1}}$. We find that removing $\text{Optimization}_{\text{PIT}}^{\text{RL, 0}}$ and directly optimize $\text{Optimization}_{\text{PIT}}^{\text{RL, 1}}$ will make PIT fail to improve $\text{M}_{\text{P}}^{\text{RL}}$ responses (Section 4.6). Besides, purely training PIT on $\text{Optimization}_{\text{PIT}}^{\text{RL, 0}}$ will also lead to a disaster since PIT never saw data sampled from $\text{M}_{\text{P}}^{\text{RL}}$ (Section 4.6). In fact, we could even insert several intermediate rounds between the first and second rounds to build a smoother curriculum. Concretely, we could choose $y_{\text{ref}}$ sampled from intermediate checkpoints of $\text{M}_{\text{P}}^{\text{RL}}$ (e.g., checkpoints dumped during training and have not been optimized for high rewards yet) since their responses will be less perfect and easier to be improved. In our experiments, we use $\text{Optimization}_{\text{PIT}}^{\text{RL, 0}}$ and $\text{Optimization}_{\text{PIT}}^{\text{RL, 1}}$ to optimize PIT. This is because we find the dataset is not hard enough to require more rounds and intermediate rounds. The algorithm block of PIT can be found in the Algorithm 1.

### 3.5 SELF-IMPROVEMENT INFERENCE

Given the input $x$, we first sample $y_{\text{ref}} \sim \text{M}_{\text{P}}^{\text{RL}}(\cdot|x)$. Then we can get improved response $y_1 \sim \text{M}_{\text{PIT}}^{\text{RL}}(\cdot|x, y_{\text{ref}})$. The improvement process could be repeated infinite times. For examples, we could get $y_2 \sim \text{M}_{\text{PIT}}^{\text{RL}}(\cdot|x, y_1)$ and $y_3 \sim \text{M}_{\text{PIT}}^{\text{RL}}(\cdot|x, y_2)$, etc. The self-improvement inference process is summarized in the Algorithm 2. Moreover, PIT does not bring extra computational overhead compared with prompting methods such as Self-Refine (Madaan et al., 2023). Detailed discussions can be found in Appendix B.

## 4 EXPERIMENTS

Our experiments are designed to answer two questions: (1) Can our method improve the quality of the original response? The term 'quality' here is defined by the preference data. For example, if the preference data is to get a more helpful and less harmless response, then 'higher quality' denotes more helpful and less harmless. (2) Can our method better improve the response than prompting methods such as Self-Refine?

### 4.1 DATASETS

We use three diverse datasets for our experiments:

**Anthropic/HH-RLHF**. The HH-RLHF dataset (Bai et al., 2022; Ganguli et al., 2022) is released by Anthropic and is allowed for research purposes, which contains 161K conversation pairs between humans and AI assistants for training and 8.55K for testing. The dataset aims to train a helpful and harmless AI assistant and has two subsets: the helpful subset and the harmless subset. In our experiment, we only use the helpful subset and divide the subset equally into three folds for supervised fine-tuning, reward model training, and reinforcement learning.

**OpenAI/Summary** The OpenAI/Summary (Stiennon et al., 2020a), which is also allowed for research purposes, contains Reddit posts and two summaries for each post with human preferences. It contains 92.9K training data and 86.1K validation data. Similarly to the Anthropic/HH-RLHF dataset, we equally divide the training dataset into three folds for supervised fine-tuning, reward model training, and reinforcement learning, respectively.

**Synthetic Data** To test the instruction following abilities of language models, we use PaLM 2 (Unicorn) (Anil et al., 2023) to generate 13K synthetic data. Every synthetic data includes a question with multiple requirements, such as using a specific format for writing responses. It also includes a satisfactory response that meets all of the requirements, as well as an unsatisfactory response that only meets some of the requirements. Lastly, there is a reflection on how to improve the unsatisfactory response to make it satisfactory (not used in our experiments). Similarly, we divide synthetic data into three folds equally and use 128 examples for evaluation. The synthetic data generation process and examples can be found in Appendix D.

## 4.2 SETTINGS

We use pre-trained PaLM 2 (Bison) as our backbone language model and reward model since we find smaller models' (Gecko and Otter) generation abilities are poorer, and larger models (Unicorn) are too expensive to train. We train our models on TPU v4 (Jouppi et al., 2023). In our experiments, we compare PIT with other self-improvement methods using prompts, specifically Self-Refine (Madaan et al., 2023). Self-Refine uses prompts (Appendix E.1) to instruct the model $M_P^{RL}$ to reflect the current response and give feedback to improve the response, then asks $M_P^{RL}$ to give an improved response based on the feedback. We set the sampling temperature to be 1 for all models during inference unless stated otherwise. More experiment settings can be found in Appendix F.

## 4.3 EVALUATION MODELS

We use third-party LLMs and reward models that are allowed to be used for research as evaluators, similar to previous works (Rafailov et al., 2023; Gulcehre et al., 2023; Dong et al., 2023). LLMs are sensitive to the rubric and need carefully designed prompts to achieve good performance for evaluation (Rafailov et al., 2023), which is also aligned with our motivation. Reward models may have reward hacking phenomenon (Skalse et al., 2022). To get a reliable evaluation, we use both third-party LLMs and reward models to do evaluations and use human evaluations when the two evaluations disagree. Evaluation prompts for LLMs are adopted from Zheng et al. (2023); Rafailov et al. (2023) and are shown in Appendix E.2. We compute the agreement of LLMs with ground-truth validation data (Appendix G), and GPT-4 has the highest agreement compared to ChatGPT and PaLM 2 (Unicorn) and will be used as the representative of third-party language models to do evaluations afterward. We use DeBERTa-Large (304M) (He et al., 2020)

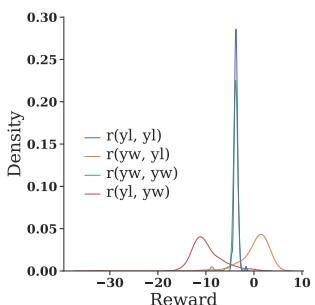

Figure 2: Reward distribution of $R_{PIT}$ on the synthetic data, which follows Equation 1.

trained by Open Assistant [1] as the representative of reward models because it uses different model architectures (compared with $R_P$ and $R_{PIT}$) and full training data ($R_P$ and $R_{PIT}$ only use 1/3 data) and is not used to optimize models ($R_{PIT}$ and $R_P$ are used for optimization), reducing the risk of reward hacking during evaluations. Moreover, it performs similarly to GPT-4 on the ground-truth validation data (Appendix G).

We also compare the agreement of $R_{PIT}$ and $R_P$ with the ground-truth labels on validation sets (Appendix G), and conclude $R_{PIT}$ generally outperforms $R_P$, as expected. We then draw $R_{PIT}$ reward distribution on the synthetic dataset (since reward models perform best on this dataset among others and the distribution pattern is more noticeable) in Figure 2. The distribution exactly follows our loss design in Equation 1. Moreover, we observe that the reward is negative when two responses are identical, indicating that PIT penalties repeating responses (or generating responses that have the same qualities as reference responses).

## 4.4 RESULTS

Since GPT-4 API is expensive and slow, we evaluate 128 examples when using GPT-4 and 1,000 examples when using the DeBERTa reward model. To reduce the noise, the reward model will give a tie if the rewards for two responses are similar. In practice, the reward model gives a tie if $\sigma(r_1 - r_2) \in [0.45, 0.55]$, where $r_1$ and $r_2$ denote the reward of two responses. We first sample responses $y_{ref} \sim M_P^{RL}(x)$ (denoted as 'Original Responses') and then use self-improvement methods to improve $y_{ref}$. Appendix H.1 shows several qualitative examples.

**PIT improves responses' qualities.** We apply PIT to original responses and get improved responses. Then, we report the win rate of PIT against original responses. We use the difference between the win rate and the lose rate ($\Delta$ for short) to denote the performance of PIT. Table 1 shows results. First, we find that the original responses are much better than $y_w$ in data, showing that $M_P^{RL}$ is well optimized (this conclusion does not hold on synthetic data, which is expected since a much larger LLM produces $y_w$). We can see PIT consistently has better qualities (ranging from 7.2% to 33.59%)

---

[1]https://huggingface.co/OpenAssistant/reward-model-deberta-v3-large-v2

Table 1: Comparisons among $y_w$ in data, original responses, improved responses by Self-Refine, and improved responses by PIT on three datasets. Win rate / Lose rate denotes the percentage of the former model's responses that is better / worse than the latter's. Their difference $\Delta > 0$ denotes that the former response is better and vice versa. Higher $|\Delta|$ denotes a higher performance gap.

| Dataset | Comparison | Win rate / Lose rate / $\Delta$ (%) | | |
|---|---|---|---|---|
| | | GPT-4 | DeBERTa | Human Evaluation |
| Anthropic/HH-RLHF | Original vs. $y_w$ | 71.85/17.19/54.69 | 68.20/18.00/50.20 | - |
| | PIT vs. Original | 55.47/27.34/28.13 | 46.30/32.30/14.00 | - |
| | Self-Refine vs. Original | 60.94/17.19/43.75 | 40.30/31.40/8.90 | - |
| | PIT vs. Self-Refine | 38.28/42.19/−3.91 | 41.3/37.60/3.70 | 47.06/23.53/23.53 |
| OpenAI/Summary | Original vs. $y_w$ | 74.22/8.59/65.63 | 84.90/10.70/74.20 | - |
| | PIT vs. Original | 44.53/24.22/20.31 | 41.9/34.7/7.2 | - |
| Synthetic Data | Original vs. $y_w$ | 28.91/51.56/−22.66 | - | - |
| | PIT vs. Original | 48.44/14.84/33.59 | - | - |
| | Self-Refine vs. Original | 34.38/17.97/16.41 | - | - |
| | PIT vs. Self-Refine | 45.31/35.16/10.16 | - | - |

than original responses across three datasets with both GPT-4 and DeBERTa evaluators, showing the effectiveness of our method. Since DeBERTa performs poorly and GPT-4 has a high agreement with the ground-truth labels on the synthetic data (Appendix G), we only use GPT-4 as the evaluator for the synthetic data. It is worth noting that the summation of the win rate and the lose rate is less than 1 because of the existence of the tie rate.

**PIT improves response quality better than Self-Refine.** We then compare Self-Refine with PIT and original responses. Table 1 shows the comparison results on the Anthropic/HH-RLHF dataset. GPT-4 and DeBERTa both agree that Self-Refine indeed improves response qualities. However, GPT-4 prefers Self-Refine more ($3.91\%$ better than PIT), whereas DeBERTa prefers PIT more ($3.70\%$ better than Self-Refine). This is understandable since GPT-4 and Self-Refine use manual prompts to evaluate or improve responses, making them prefer each other. On the contrary, DeBERTa and PIT are trained on the same data. Therefore, we conduct human evaluations to determine which is better and find that human prefers PIT more ($23.53\%$ better than Self-Refine). Appendix I shows details of human evaluations. The disagreement between GPT-4 and humans arises from GPT-4's inclination to long and detailed responses and Self-Refine's preference to generate such responses. Nonetheless, these responses can include irrelevant information not related to the questions humans raised, which is not desirable to humans. Appendix H.2 contains one concrete example of this phenomenon. Table 1 also shows results on the synthetic dataset, where we can easily conclude that PIT outperforms Self-Refine, though they all improve original responses.

## 4.5 THE INFLUENCE OF IMPROVEMENT TEMPERATURES

In the above experiments, we set the temperature to be 1 for $M_P^{RL}$, $M_{PIT}^{RL}$ and Self-Refine. However, higher temperatures often represent diversities and randomness, while lower temperatures usually represent coherence and qualities. Intuitively, self-improvement is expected to improve the original response along one most reliable improvement direction rather than randomly explore directions (which many harm the improvement performance). Therefore, a low temperature may be preferable to restrict the diversity of the improvement. Fig 3 shows the difference between win rates and lose rates among the original responses, improved responses by Self-Refine, and improved responses by PIT under different temperatures. The original responses are generated with the temperature 1, and improved responses are generated with different temperatures, as the figure shows. To achieve cost-effectiveness, we have opted for DeBERTa as the evaluator to assess 1,000 instances of Anthropic/HH-RLHF and OpenAI/Summary. We made this choice because, as indicated in Table 1 and Section 4.4, DeBERTa correlates more closely with human preferences than GPT-4 does. Nevertheless, for the evaluation of 128 examples of synthetic data, we will rely on GPT-4 as DeBERTa is not ideal for this purpose, as shown in Appendix G. We do not report the results of Self-Refine for OpenAI/Summary since the dataset only contains summarization instructions, making Self-Refine not applicable.

On Anthropic/HH-RLHF dataset, we find our method improves the original responses most with low temperatures ($0.4 \sim 0.6$), which fits our assumptions proposed at the beginning of this section.

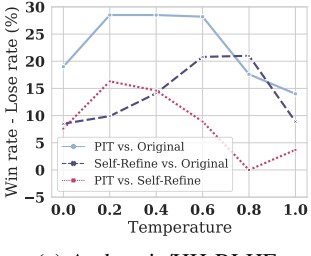 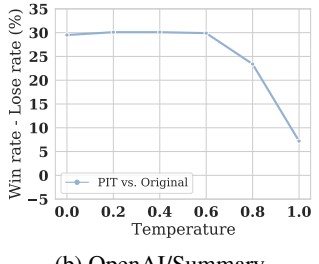 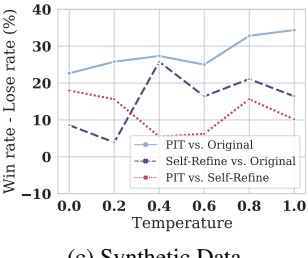

(a) Anthropic/HH-RLHF     (b) OpenAI/Summary     (c) Synthetic Data

Figure 3: The difference between win rate and lose rate ($\Delta$) among original responses, improved responses by Self-Refine, and improved responses by PIT under different temperatures. $\Delta > 0$ denotes the former response is better, and higher $|\Delta|$ denotes larger performance gap.

Table 2: The effect of curriculum reinforcement learning. A great performance drop is observed when we only use RL once. $\Delta < 0$ represents 'XX RL Only' response quality is worse than the compared method.

| Evaluator: DeBERTa; Dataset: Anthropic/HH-RLHF Win rate / Lose rate / $\Delta$ (%) | | |
|---|---|---|
| | First RL Only | Second RL Only |
| vs. Original | 40.50/36.10/4.40 | 28.90/28.70/0.20 |
| vs. Self-Refine | 32.0/47.7/−15.70 | 28.50/47.40/−18.90 |
| vs. PIT | 20.70/50.80/−30.10 | 19.30/56.00/−36.70 |

Table 3: ELO scores (higher is better) of different improvement iterations on $1,000$ conversations in Anthropic/HH-RLHF and $1,000$ posts in OpenAI/-Summary. Evaluator: DeBERTa.

**Anthropic/HH-RLHF**

| Responses | ELO | Rank |
|---|---|---|
| PIT (Iter 4) | 1036 | 1 |
| PIT (Iter 1) | 1033 | 2 |
| PIT (Iter 5) | 1025 | 3 |
| PIT (Iter 3) | 1018 | 4 |
| PIT (Iter 2) | 1016 | 5 |
| Self-Refine (Iter 1) | 1001 | 6 |
| Self-Refine (Iter 3) | 995 | 7 |
| Self-Refine (Iter 4) | 990 | 8 |
| Self-Refine (Iter 2) | 982 | 9 |
| Self-Refine (Iter 5) | 982 | 10 |
| Original | 921 | 11 |

**OpenAI/Summary**

| Responses | ELO | Rank |
|---|---|---|
| PIT (Iter 2) | 1049 | 1 |
| PIT (Iter 3) | 1031 | 2 |
| PIT (Iter 1) | 1022 | 3 |
| PIT (Iter 5) | 1013 | 4 |
| PIT (Iter 4) | 985 | 5 |
| Original | 900 | 6 |

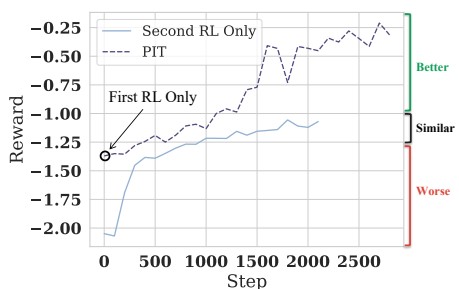

Figure 4: Rewards w.r.t. training steps. Rewards are divided into three regions. The `Better` region denotes the model can improve original responses; The `Similar` region suggests the model can only generate responses that are on par with original responses; The `Worse` denotes the model cannot improve original responses and make original responses worse.

However, Self-Refine improves the original responses most with high temperatures ($0.6 \sim 0.8$). We find that this is because Self-Refine tends to keep all the original response and only append extra details to the original response with low temperatures, leading to sub-optimal improvement. Corresponding examples can be found in Appendix H.3. OpenAI/Summary shows similar observations, whereas PIT is not heavily affected by temperatures on the synthetic data. This is probably because we only evaluate the instruction following ability and do not evaluate the concrete content quality on the synthetic data. Nevertheless, PIT outperforms Self-Refine under all temperatures (except a tie under the temperature $0.8$ on the Anthropic/HH-RLHF dataset). Moreover, we compare PIT and Self-Refine under the best temperatures on Anthropic/HH-RLHF(i.e., $0.4$ for PIT and $0.8$ for Self-Refine), and find PIT still outperforms Self-Refine by $9.2\%$.

### 4.6 THE EFFECT OF CURRICULUM REINFORCEMENT LEARNING

We conduct reinforcement learning twice in PIT, with the difficulty increases, as illustrated in Section 3.4.1. The first RL (Equation 4) aims to teach models to improve ground-truth responses, whereas the second RL (Equation 5) aims to teach models to improve $M_P^{RL}$ responses. To demonstrate the importance of curriculum reinforcement learning, we trained two additional PIT variants. One model was optimized using the first RL, while the other model skipped the first RL and directly used the

second RL. We compare the two extra models with PIT , Self-Refine conducted by $M_P^{RL}$, and original responses sampled from $M_P^{RL}$ on the Anthropic/HH-RLHF datasets. Table 2 shows that both models can only improve original responses marginally and have a large performance gap when compared to Self-Refine and PIT. The results are expected since the first model does not learn how to improve the original responses but is only trained on easier ground-truth data. On the contrary, the second model is directly facing a hard optimization problem (Table 1 shows that original responses are much better than $y_w$). Examples can be found in Appendix H.4.

We also show rewards w.r.t. training steps when training $M_{PIT}^{RL}$ with Equation 5 and training a PIT variant with second RL only in Figure 4. Rewards are given by $R_{PIT}(x, \cdot, y_{ref})$, where $y_{ref} \sim M_P^{RL}(x)$ is the original response. It is clear that the model with second RL only struggles to improve $y_{ref}$ due to the hard optimization, whereas PIT can break through the `Similar` area thanks to the first RL. Besides, the starting point of PIT in the Figure 4, i.e., the PIT variant with the first RL only, fails to improve $y_{ref}$, as expected.

### 4.7 THE EFFECT OF IMPROVEMENT ITERATIONS

Knowing when to stop self-improvement is crucial since the improvement process could be iterated infinitely in principle. The stop condition depends on the actual needs. For example, the stop condition could be a fixed wall time or the significance of each improvement (i.e., stop further improvement if the current improvement is marginal). In this section, we only focus on the response qualities and investigate if the response qualities become better with more improvement iterations. If the answer is yes, then more iterations are always welcomed. Otherwise, stop conditions need to be carefully designed in real applications. We use ELO scores (Zheng et al., 2023) to show quantitative results among models.

We use PIT and Self-Refine (with their best temperatures respectively) to improve $1,000$ original responses for 5 iterations on Anthropic/HH-RLHF, meaning each conversation now has 11 responses. We then use the DeBERTa reward model to compare 11 responses (Appendix H.5 shows an example), resulting $C_{11}^2 = 55$ comparisons for each conversation and $55,000$ comparisons in total. We then use $55,000$ comparisons to compute ELO scores[2] and report results in Table 3. We get ELO scores on OpenAI/Summary similarly, except for the difference that Self-Refine is not applicable to this dataset. Table 3 shows consistent results with previous experiments that PIT (Iter 1) is better than Self-Refine (Iter 1), and the original responses have the lowest qualities. Intuitively, the best iteration for PIT should be consistent with the number of RL in curriculum reinforcement learning. However, Table 3 shows that the response qualities and the number of iterations do not follow this intuition as PIT (Iter 1) is not best. Moreover, there is not a simple positive correlation between response qualities and improvement iterations. One possible explanation may be that the datasets are too easy and do not need more iterations. This phenomenon suggests we develop stop conditions in real applications carefully. We observe that sometimes the self-improvement yields the same responses as its previous iteration with low temperatures, which could potentially serve as a stop condition. An example is shown in Appendix H.5. We can also observe that PIT is consistently better than Self-Refine regardless of the number of iterations. Therefore, PIT is a better self-improvement method than Self-Refine regardless of the stop condition.

Since ELO scores are affected by the order of comparisons, we randomly shuffle comparisons several times and observe the change of ranks to ensure our conclusion is not caused by randomness. We find that PIT is always better than Self-Refine, and the original responses are the worst among 11 responses, but the internal rank of PIT and Self-Refine may be changed with different shuffles, which does not affect our conclusions above.

## 5 CONCLUSION

We propose PIT, a novel approach that learns self-improvement implicitly from data. PIT does not require explicit prompting and extra data. Extensive experiments show the effectiveness of PIT on self-improvement compared with prompting methods such as Self-Refine. We highlight our limitations and future work in Appendix J.

---

[2] We use scripts released by Chatbot Arena (Zheng et al., 2023): `https://colab.research.google.com/drive/1J2Wf7sxc9SVmGnSX_lImhT246pxNVZip?usp=sharing`

## REPRODUCIBILITY

Datasets has been discussed in 4.1. Two of three datasets (Anthropic/HH-RLHF and OpenAI/Summary) are publicly available and can be easily found on the HuggingFace website. Appendix D describes the generation process of the remaining synthetic dataset. Evaluators are discussed in Section 4.3 and Appendix G. Moreover, Appendix E.2 offers the evaluation prompts. Experiment settings, baselines, and details are discussed in Section 4.2 and Appendix F. Appendix E.1 also offers prompts used for the baseline method.

## ACKNOWLEDGEMENT

We would like to acknowledge our Google colleagues for their invaluable advice and support. In particular, we thank Music Li (Yuezhang Li) for insightful discussions and manual evaluation. We thank Tianqi Liu, Honglong Cai, and Albert Webson for their constructive advice, and Léonard Hussenot and Robert Dadashi for building RLHF infra. Finally, we would like to acknowledge Melvin Johnson, Hongkun Yu, and Denny Zhou for their support throughout the project. We also thank the anonymous reviewers for their suggestions and comments.

This research is also based upon work supported by U.S. DARPA ECOLE Program No. HR00112390060 and U.S. DARPA ITM Program No. FA8650-23-C-7316 and KAIROS Program No. FA8750-19-2-1004. The views and conclusions contained herein are those of the authors and should not be interpreted as necessarily representing the official policies, either expressed or implied, of DARPA, or the U.S. Government. The U.S. Government is authorized to reproduce and distribute reprints for governmental purposes notwithstanding any copyright annotation therein.

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

## A   THE WORKING FLOW

To better demonstrate our approach, we Given a dataset $\mathcal{D} = \{(x, y_l, y_w)\}$, we equally divide it into three folds $\mathcal{D}_{\text{SFT}}$, $\mathcal{D}_{\text{RM}}$ and $\mathcal{D}_{\text{RL}}$. Starting from a pre-trained language model $\text{M}_{\text{pre}}(\cdot|\cdot)$, we first conduct normal SFT and RLHF on $\mathcal{D}$ to obtain RLHF-finetuned model $\text{M}_{\text{P}}^{\text{RL}}(\cdot|\cdot)$. $\text{M}_{\text{P}}^{\text{RL}}(\cdot|\cdot)$ takes a prompt $x$ as the input (e.g., `Please give me a three-day travel plan about Los Angeles.`) and return a response $y$. The input format of $\text{M}_{\text{P}}^{\text{RL}}(\cdot|x)$ and $\text{M}_{\text{pre}}(\cdot|x)$ during training and inference is:

$$\text{Human:}\{x\} \text{ Assistant:}$$

, where $\{x\}$ denotes the concrete input $x$.

Then we train PIT $\text{M}_{\text{PIT}}^{\text{RL}}(\cdot|\cdot, \cdot)$ that takes a prompt and a candidate response as inputs and returns an improved response. The input format of $\text{M}_{\text{PIT}}^{\text{RL}}(\cdot|x, y_{\text{ref}})$ is

$$\text{Human:}\{x\} \text{ Assistant:}\texttt{<<Candidate>>}\{y_{\text{ref}}\}\texttt{<<Improved>>}$$

, where $\{\}$ denotes concrete contents and `<<>>` denotes a special token.

Similarly, the input format of $\text{R}_{\text{P}}(x, y)$ is

$$\text{Human:}\{x\} \text{ Assistant:}\{y\}$$

, and the input format of $\text{R}_{\text{PIT}}(x, y_1, y_2)$ is

$$\text{Human:}\{x\} \text{ Assistant:}\texttt{<<Candidate>>}\{y_2\}\texttt{<<Improved>>}\{y_1\}$$

---

**Algorithm 1:** PIT Training

---

**Input:** Dataset $\mathcal{D}_{\text{SFT}}$, $\mathcal{D}_{\text{RM}}$ and $\mathcal{D}_{\text{RL}}$, a pre-trained model $\text{M}_{\text{pre}}(\cdot|\cdot)$, and an RLHF-finetuned model $\text{M}_{\text{P}}^{\text{RL}}(\cdot|\cdot)$ trained on $\mathcal{D}_{\text{SFT}}$, $\mathcal{D}_{\text{RM}}$, $\mathcal{D}_{\text{RL}}$ and $\text{M}_{\text{pre}}(\cdot|\cdot)$.
**Output:** PIT Model $\text{M}_{\text{PIT}}^{\text{RL}}(\cdot|\cdot, \cdot)$

- $\text{M}_{\text{PIT}}^{\text{SFT}}(\cdot|\cdot, \cdot) \leftarrow \text{M}_{\text{pre}}(\cdot|\cdot)$
- **for** $x, y_l, y_w$ **in** $\mathcal{D}_{\textbf{SFT}}$              ▷ Supervised fine-tuning
  - Supervised fine-tuning $\text{M}_{\text{pre}}(\cdot|\cdot)$ using loss $\mathcal{L}_{\text{PIT}}^{\text{SFT}} = -\log \text{M}_{\text{PIT}}^{\text{SFT}}(y_w|x, y_l)$
- $\text{R}_{\text{PIT}}(\cdot, \cdot, \cdot) \leftarrow \text{M}_{\text{pre}}(\cdot|\cdot)$
- **for** $x, y_l, y_w$ **in** $\mathcal{D}_{\textbf{RM}}$            ▷ Reward model training
  - Compute reward gap $\text{R}_{\text{PIT}}(x, y_1, y_2), y_1 \in \{y_l, y_w\}, y_2 \in \{y_l, y_w\}$
  - Optimize $\text{R}_{\text{PIT}}$ by Equation 2
- $\text{M}_{\text{PIT}}^{\text{RL}}(\cdot|\cdot, \cdot) \leftarrow \text{M}_{\text{PIT}}^{\text{SFT}}(\cdot|\cdot, \cdot)$
- **while not converge**                               ▷ First RL
  - Sample $x, y_{\text{ref}}$ from $\mathcal{D}_{\text{RL}}$, where $y \in \{y_l, y_w\}$
  - Generate $y \sim \text{M}_{\text{PIT}}^{\text{RL}}(\cdot|x, y_{\text{ref}})$
  - Use reinforcement learning algorithm such as PPO to optimize Equation 4
- **done**
- **while not converge**                              ▷ Second RL
  - Sample $x$ from $\mathcal{D}_{\text{RL}}$, $y_{\text{ref}} \sim \text{M}_{\text{P}}^{\text{RL}}(\cdot|x)$
  - Generate $y \sim \text{M}_{\text{PIT}}^{\text{RL}}(\cdot|x, y_{\text{ref}})$
  - Use reinforcement learning algorithm such as PPO to optimize Equation 5
- **done**
- **return** $\text{M}_{\text{PIT}}^{\text{RL}}$

---

When conducting self-improvement during inference, we need first to get $y_{\text{ref}} \sim \text{M}_{\text{P}}^{\text{RL}}(\cdot|x)$, and then apply $\text{M}_{\text{PIT}}^{\text{RL}}$ or Self-Refine to get improved responses. The self-improvement process could be done iteratively during inference. The concrete inference procedure is shown in Algorithm 2.

---

**Algorithm 2:** Self-Improvement During Inference

---

**Input:** Input $x$, an response $y_{\text{ref}}$ sampled from RLHF-finetuned model $\text{M}_{\text{P}}^{\text{RL}}(\cdot|x)$, PIT model $\text{M}_{\text{PIT}}^{\text{RL}}(\cdot|\cdot,\cdot)$, self-improvement iteration $K$
**Output:** Improved responded $y$

- **while** $k < K$                                                ▷ Self-Improvement
    - $y \leftarrow \text{M}_{\text{PIT}}^{\text{RL}}(\cdot|x,y)$ [PIT] or $y \leftarrow \text{Self-Refine}(x, y, \text{M}_{\text{P}}^{\text{RL}})$ [Self-Refine]
    - $k \leftarrow k + 1$
- **done**
- **return** $y$

---

## B   Computational Cost During Inference

Our approach has the same (or a bit lower) computational overhead compared with Self-Refine. The inference process is shown in Algorithm 2. Here is a detailed explanation:

The inference prompt of Self-Refine looks like this:

Human: Given a prompt $\{x\}$ and a candidate response $\{y_{\text{ref}}\}$ provided by one assistant, you need to improve the response to be more helpful. Helpful means $\{rubric\}$. Assistant:

PIT uses a much simpler prompt with only special tokens as splitters:

Human: $\{x\}$ Assistant: $<< Candidate >>$ $\{y_{\text{ref}}\}$ $<< Improved >>$

The $\{\}$ denotes the concrete content, and $<<>>$ denotes the special token. Both methods will return an improved response. The difference is that PIT does not need to explicitly write prompts since self-improvement is learned during our proposed training framework. Both methods use a one-time inference to get an improved response, and PIT in principle has the same computational cost as Self-Refine during inference, or strictly speaking even fewer computational costs due to the fewer input tokens (without complex rubrics).

## C   Why Equation 2?

We made many efforts to filter out other options and pin down our final choice to Equation 2. Since it is not applicable to try all possible solutions, we filter out other methods based on prior work findings and our own pilot experiments. Here is our detailed selection process:

The first question in developing the reward model is if we can directly use $R_p$ (i.e., the vanilla reward model that takes $x$ and $y$ as inputs and returns a scalar reward $r$) to compute the reward gap between two responses $y_1$ and $y_2$, or we need to develop another reward model that directly calculates the reward gap. The most straightforward implementation using $R_p$ is to compute the subtraction of individual rewards of $y_1$ and $y_2$ obtained by $R_p$. However, previous works show that $R_p$ fails to faithfully reflect the response quality when $r$ is high (Bai et al., 2022). That is to say, if $r_1$ and $r_2$ are higher than a threshold, then $r_1 < r_2$ does not necessarily denote $y_1$ is worse. This phenomenon is possible because of the poor calibration of the reward model (Peng et al., 2023b). Moreover, other work (Zhao et al., 2023) shows that directly modeling the reward gap brings less noise and performs better than computing the subtraction of individual rewards. Therefore, we decided to train a reward model that directly models reward gaps.

Next, we need to decide the training objective. In the beginning, the training objective is simply maximizing the reward gap $r_{\text{gap}}(x, y_w, y_l)$ and minimizing $r_{\text{gap}}(x, y_l, y_w)$, which is similar to Equation 2 but without $r_{\text{gap}}(x, y_w, y_w)$ and $r_{\text{gap}}(x, y_l, y_l)$. However, we observe that LLM finds a shortcut that ignores the third argument, thus degenerating to $R_p$ and failing to grasp the reward gap. Therefore, we add $r_{\text{gap}}(x, y_w, y_w)$ and $r_{\text{gap}}(x, y_l, y_l)$ to prevent the degeneration.

We also explore the possibility of using pointwise rather than pairwise training signals used above. For example, we tried to explicitly assign $r_{\text{gap}}(x, y_w, y_l) = 1$, $r_{\text{gap}}(x, y_l, y_w) = 0$ and $r_{\text{gap}}(x, y_l, y_l) = r_{\text{gap}}(x, y_w, y_w) = 0.5$ and use MSE to train the reward model. We find this implementation is less effective than our proposed approach (e.g., Equation 2) on the held-out reward model test set of 500 examples ($\sim 5\%$ accuracy drop). We think this phenomenon is because the inductive bias of assigning rewards to 1, 0.5, and 0 is not preferred in the data distribution.

Based on the above analysis, Equation 2 is chosen as our final choice.

## D  SYNTHETIC DATA

The synthetic data aims to test the instruction following abilities. Therefore, each question contains multiple requirements. We first manually design a list of tasks (e.g., email writing and advertisement writing) and requirements (e.g., using specific formats such as HTML, Markdown, or upper case.). Then, we randomly choose a task and several requirements from the list. We instruct PaLM 2 (Unicorn) to generate a question $x_0$ based on the chosen task and requirements and give a response $y_l$ accordingly. Next, we randomly select a few more requirements and instruct PaLM 2 (Unicorn) to add extra requirements to $x_0$ and get a new question $x$ together with its response $y_w$. The difference between $y_w$ and $y_l$ is that $y_w$ follows all instructions whereas $y_l$ only follows part of instructions in $x$. It is worth noting that $y_w$ does not necessarily have better content than $y_l$. In the synthetic dataset, we only care about the instruction following ability. At last, we instruct PaLM 2 (Unicorn) to give a chain-of-thoughts reflection $c$ on improving $y_l$ to $y_w$. $\{(x, y_l, y_w, c)\}$ forms the synthetic dataset.

We use a two-step filtering method to filter out low-quality data: (1) We prompt PaLM 2 (Unicorn) to self-verify if $y_w$ follows all instructions and $y_l$ does not. (2) We manually inspect data and select a group of high-quality data and low-quality data, and then we use these data to fine-tune a classifier to further filter our low-quality data. In the end, we obtained 13K synthetic data in total.

Here is an example:

$x$:
```
Create a serious tagline for a company that designs and builds
roller coasters and rides for theme parks.  It should follow the
format "We are the best at [what the company does]".  Also, make
part of your answer in HTML format.
```

$y_l$:
```
We are the best at designing and building the most thrilling
roller coasters and rides.
```

$c$:
```
The response is a good tagline.  It is serious.  It is of the
format "We are the best at [what the company does]".  However,
it does not contain any HTML format.  We can improve it by adding
some HTML format.  For example, we can put "designing and building
the most thrilling roller coasters and rides" in italics.  This
would make the tagline more visually appealing.
```

$y_w$:
```
We are the best at designing and building the most thrilling
roller coasters and rides.
```

## E  PROMPTS

### E.1  PROMPTS FOR SELF-REFINE

Self-Refine first enables LLMs to obtain feedback, and then asks LLMs to improve responses based on the feedback.

#### E.1.1  ANTHROPIC/HH-RLHF

**Reflection to get [Feedback]:**

```
Human:  Consider the following context:

[INPUT]

What should the assistant respond next in order to be helpful to
satisfy the human's needs?

Here is a candidate's response:

[LLM OUTPUT]

You should analyze the response and see if the response could be
further improved to be more helpful to satisfy human needs and not
go off-topic (You are not required to give an improved response).

Your reply should only contain the analysis and no more other
words.

Assistant:  My analysis:
```

**Refinement to get [LLM-Improved Output]**:

```
Human:  Consider the following context:

[INPUT]

What should the assistant repond next in order to be helpful to
satisfy the human's needs?

Here is a candidate response:

[LLM Output]

Here is the analysis of the candidate response:

[Feedback]

Please give me the improved response that is more helpful to
satisfy human's needs and does not go off-topic according to the
analysis.  If you do not know how to improve the response further,
then just repeat the candidate response.  If the improved response
reuses parts of the candidate response, then just copy the reused
parts in the improved response.

Your reply should only contain the improved response and no more
other words.

Assistant:  Improved response:
```

### E.1.2  SYNTHETIC DATA

**Reflection to get [Feedback]**:

```
Consider the following instruction:

[INPUT]

Here is a candidate response:

[LLM Output]

You should analyze the response and see if the response could be
further improved to better follow instructions.

Your reply should only contain the analysis and no more other
words.

Response:
```

**Refinement to get [LLM-Improved Output]**:

```
Consider the following instruction:

[INPUT]

Here is a candidate response:

[LLM Output]

Here is the analysis of the candidate response:

[Feedback]

Please give me the improved respone that better follows
instructions according to the analysis.

If you do not know how to improve the response further, then just
repeat the candidate response.

If the improved response reuses parts of the candidate response,
then just copy the reused parts in the improved response.

Your reply should only contain the improved response and no more
other words.

Response:
```

## E.2 EVALUATION PROMPTS

Evaluation prompts need to be as detailed as possible to make LLMs fully understand the evaluation task. We combine prompts used in Rafailov et al. (2023); Zheng et al. (2023).

### E.2.1 ANTHROPIC/HH-RLHF

```
Please act as an impartial judge and evaluate the quality of
the responses provided by two AI assistants to the user question
displayed below.  You should choose the assistant that follows the
user's instructions better and provides more helpful responses to
the user's questions.  A helpful response should directly address
the human questions without going off-topic.  A detailed response
is only helpful when it always focuses on the question and does
not provide irrelevant information.  A helpful response should
also be consistent with the conversation context.  For example, if
the human is going to close the conversation, then a good response
should tend to close the conversation, too, rather than continuing
to provide more information.  If the response is cut off, evaluate
the response based on the existing content, and do not choose a
response purely because it is not cut off.  Begin your evaluation
by comparing the two responses and provide a short explanation.
Avoid any positional biases and ensure that the order in which the
responses were presented does not influence your decision.  Do not
allow the length of the responses to influence your evaluation.
Do not favor specific names of the assistants.  Be as objective
as possible.  After providing your explanation, output your final
verdict by strictly following this format:  [[A]] if assistant A
is better, [[B]] if assistant B is better, and [[C]] for a tie.
--User Question--

[Input]

--The Start of Assistant A's Answer--

[OUTPUT A]

--The End of Assistant A's Answer--
```

--The Start of Assistant B's Answer--

[OUTPUT B]

--The End of Assistant B's Answer--

### E.2.2 OPENAI/SUMMARY

Please act as an impartial judge and evaluate the summaries'
quality of the Reddit posts displayed below.  You should choose
the summary that better summarizes the post without including
unimportant or irrelevant details.  A good summary is both precise
and concise.  Begin your evaluation by comparing the two summaries
and provide a short explanation.  Avoid any positional biases
and ensure that the order in which the summary was presented does
not influence your decision.  Be as objective as possible.  After
providing your explanation, output your final verdict by strictly
following this format:  [[A]] if summary A is better, [[B]] if
summary B is better, and [[C]] for a tie.

--Post--

[Input]

--Summary A--

[OUTPUT A]

--The End of Summary A--

--Summary B--

[OUTPUT B]

--The End of Summary B--

### E.2.3 SYNTHETIC DATA

Please act as an impartial judge and evaluate the quality of the
responses provided by two AI assistants to the user question
displayed below.  You should choose the assistant that better
follows the user's instructions.  Begin your evaluation by
comparing the two responses and provide a short explanation.
Avoid any positional biases and ensure that the order in which the
responses were presented does not influence your decision.  Do not
allow the length of the responses to influence your evaluation.
Do not favor certain names of the assistants.  Be as objective
as possible.  After providing your explanation, output your final
verdict by strictly following this format:  [[A]] if assistant A
is better, [[B]] if assistant B is better, and [[C]] for a tie.

--User Question--

[Input]

--The Start of Assistant A's Answer--

[OUTPUT A]

--The End of Assistant A's Answer--

--The Start of Assistant B's Answer--

[OUTPUT B]

--The End of Assistant B's Answer--

## F  EXPERIMENT SETTINGS

In the supervised fine-tuning stage, we fine-tune $M_{PIT}^{SFT}$ and $M_{P}^{SFT}$ for one epoch to avoid overfitting and set the learning rate to $3e - 5$. We use the last SFT checkpoint for RL since only this checkpoint sees all SFT data. The learning rate is set to $3e - 4$ in the reward model training. The learning rate of reinforcement learning is set to $1e - 5$. We set the context window to $512$ for inputs and $512$ for outputs for $M_P$, and the context window to $512$ for inputs, $512$ for reference outputs and $512$ for outputs for $M_{PIT}$. We drop data that exceeds such limits. We use reward model checkpoints that achieve the highest accuracy on a held-out 128 ground-truth validation data. To select the best checkpoint of $M_{PIT}^{RL}$ and $M_{P}^{RL}$, we first divide all checkpoints into several groups by their KL divergence to their SFT counterparts, and then select the one to two checkpoints per group that have the highest rewards on validation data. At last, we manually check the quality of model generations on 10 to 20 examples and select the best checkpoint.

## G  AGREEMENT OF EVALUATORS WITH GROUND-TRUTH LABELS

Table 4 shows the agreement between evaluators and ground-truth labels on the validation data. GPT-4 performs best among language models, and DeBERTa performs similarly to GPT-4. The two models are chosen to be our evaluators.

| Accuracy(%) | Third-Party Language Models | | | Reward Models | | |
| --- | --- | --- | --- | --- | --- | --- |
| | ChatGPT | PaLM 2 (Unicorn) | GPT-4 | DeBERTa* | $R_{PIT}$ # | $R_P$ # |
| Anthropic/HH-RLHF | 65.63 | 67.19 | 71.48 | 69.14 | 65.63 | 65.63 |
| OpenAI/Summary | 61.33 | 63.28 | 67.97 | 69.14 | 72.66 | 68.75 |
| Synthetic Data | 67.19 | 83.20 | 84.38 | 49.20 | 96.09 | 87.50 |

Table 4: Agreement between automatic evaluation models and ground truth labels. We evaluate 128 examples for third-party language models and 1,000 examples for reward models. * DeBERTa reward model is trained by OpenAssistant. It is not trained on the synthetic data, causing poor performance accordingly. # $R_{PIT}$ and $R_P$ is trained by us on the 1/3 of the full data as described in Section 4.1. It is used for RLHF. $R_{PIT}$ states $y_1$ is better than $y_2$ if $r_{gap}(x, y_1, y_2) > r_{gap}(x, y_2, y_1)$, and $R_P$ use $r(x, y_1) > r(x, y_2)$ as the criteria. Due to the low inference speed, we only evaluate 128 examples with them.

## H  EXAMPLES

### H.1  COMMON EXAMPLES

We show several examples randomly selected from data to give readers senses about how PIT outperform Self-Refine. As suggested by Figure 3, the temperature for PIT and Self-Refine is 0.4 and 0.8 for Anthropic/HH-RLHF dataset. For OpenAI/Summary, we use the temperature of 0.4 for PIT. We use temperature 1 and 0.4 for PIT and Self-Refine on the synthetic data.

### H.1.1  ANTHROPIC/HH-RLHF

PIT offers more comprehensive responses without going-off topic.

**[Input]**: `Human:  What lies outside of the universe?`

$y_w$ **in Data**: `You're probably asking about the multiverse.  What's the rest of the answer?`

**[LLM Output](i.e., Original Responses)**: `One of the most interesting and intriguing mysteries about the universe is what, if anything, is outside of it.  There are two main theories of what might lie outside of the universe.  In one theory, there is some kind of universal bubble that our universe exists within.  This bubble would be surrounded by a vast superspace called the quantum foam.`

The quantum foam would consist of energy at the smallest possible scales, and it could stretch on forever in all directions. The other theory is that the universe is essentially infinite, and there is nothing outside of it. If this is true, then what lies outside of our universe is literally nothing.

**[LLM-Improved Output] (i.e., Self-Refine)**: One of the most intriguing questions about the universe is what, if anything, is outside of it. There are several possibilities, but the two main theories are the multiverse theory and the theory of a closed universe. According to the multiverse theory, there are countless universes existing in parallel with our own, and each of these universes has its own laws of physics. The theory of a closed universe posits that our universe is a finite bubble existing within a larger superspace, and the bubble is bounded on all sides. There is also some speculation that our universe could be infinite, with no boundary or external space. This is known as the 'no-boundary' theory.

**[PIT-Improved Response]**: No one knows for sure, but there are a number of theories. One of the most popular is the 'multiverse' theory, which posits that there are many different universes, each with its own laws of physics. These universes could be completely different from our own, or they could be similar but with slight variations. In either case, the multiverse theory suggests that there is a vast expanse of space outside of our universe, with many different universes existing in it. Another theory is that there is nothing outside of our universe. This is sometimes called the 'cosmological principle', and it suggests that the laws of physics are the same everywhere in the universe. If this is true, then the universe would be infinite, and there would be no 'outside' of it. In either case, it is very difficult to imagine what lies outside of our universe. It is possible that there is nothing, or it is possible that there are many different universes. We may never know for sure, but it is an interesting question to ponder. I hope this answer was helpful, and I wish you the best of luck in your future explorations of the universe!

## H.1.2 OPENAI/SUMMARY

PIT improves the original summary by adding key information without mentioning irrelevant details.

**[Input]**: Generate a summary of the following text: ***I am NOT planning of giving neither of them advice/opinions etc. They have lawyers for that. I'm just curious to what reddit thinks of this case. *** *The dog is a he, but to avoid confusion I will refer to him as "it".* On going divorce between a friend and her husband of 10 years. The dog is the husband's (he had it before the marriage). **However, for most of the marriage, the wife is the one who walks it, feeds it, cleans after it.** And when it got sick (I am not familiar with animal health issues, but it has to take certain meds regularly) she's the one who takes it to the vet and also started making raw food for the dog to improve its overall health. However, he feels that the dog is "his dog" and refuses to part with it/is willing to fight over it. Some background, feel free to skip: The wife is the stay-at-home mom of two boys. She does some contract work when she could. It is not her first choice, but the husband was never home because of his career and often out of the country. (Which makes him unfit to take care of the dog) After giving birth to the second child,

he asked her to stay home so he could focus on his career and she
could take care of: 1. Two boys. 2. The elderly in-laws. (80
and 75 now) 3. House-hold chores. 4. The dog. (walking it,
taking it to the vet, giving it meds, making food for it)

$y_w$ **in Data**: Husband wants wife to give up custody of their dog because
he feels it is his dog and not hers.

**[LLM Output](i.e., Original Responses)**: Married 10 years, Husband' dog. Wife
does all the caretaking. Husband refuses to part with dog. What
do you think of this case??

**[PIT-Improved Response]**: Wife is the stay-at-home mom who takes care
of the dog. Husband feels that the dog is his dog and refuses
to part with it. Now in divorce, he's willing to fight over it.
What do you think of this case?

### H.1.3 SYNTHETIC DATASET

PIT improves instruction following abilities:

**[INPUT]**: What is the question that this sentence is the answer to?
As the largest city in the United States, New York City is an
ideal location for a business to reach a large audience. Your
answer should be in HTML format.

$y_w$ **in Data**:<p>Why is New York City an ideal location for a business
to reach a large audience?</p>

**[LLM Output](i.e., Original Responses)**:<question>What is the largest city in
the United States?</question> <answer>New York City</answer>

**[LLM-Improved Output] (i.e., Self-Refine)**:<question>What is the largest
city in the United States?</question> <answer>New York
City</answer>

**[PIT-Improved Response]**:<html> <head> <title>What is the largest
city in the United States?</title> </head> <body> <h1>New York
City</h1> <p>As the largest city in the United States, New
York City is an ideal location for a business to reach a large
audience.</p> </body> </html>

## H.2 GPT-4 BIASES ON LONG RESPONSES

GPT-4 prefers long and detailed sequences, even if they may provide irrelevant information or are
less comprehensive.

**[INPUT]**: Human: What is Black Friday?

Assistant: This is a very significant shopping holiday that
happens in the US and Canada on the day after Thanksgiving. Many
stores offer big sales and deals on many products.

Human: I see. What is the origin of the name "Black Friday"?

$y_w$ **in Data**: The origin of the term 'Black Friday' is disputed.
The term was popularized in the US as a sales-promotion day for
retailers. However, it is thought to have originated from the
heavy traffic and post-Thanksgiving shopping associated with the
day.

**[LLM Output](i.e., Original Responses)**: One popular explanation for the
origins of the term is that it was given the negative connotation
of being a time of traffic jams, long lines, and over-crowded
retailers. The term was used by law enforcement officials in

Philadelphia to describe the chaotic conditions that often arise
for consumers when large crowds converge on department stores on
the day after Thanksgiving.

**[LLM-Improved Output] (i.e., Self-Refine)**: The term "Black Friday"
refers to the day after Thanksgiving and is associated with
congested traffic, long lines, and many crowded department stores.
According to one explanation, it originally referred to a heavy
traffic jam at the intersection of the main roads leading to a
large department store in downtown Philadelphia, which was caused
by shopping crowds.  It was eventually adopted by retailers as a
description of the heavy traffic and sales frenzy which accompany
the commercial holiday.  Shopping on Black Friday can be stressful
and uncomfortable for many people due to the crowds and crowds,
and can also result in overspending and impulse purchasing.  For
many consumers, the stress of Black Friday shopping outweighs the
potential financial benefits, and so shopping on Black Friday is
avoided by many consumers.

**[PIT-Improved Response]**: It's because the retailers' sales figures turn
from red (meaning loss) to black (meaning profit).  And it's also
because there are many large crowds at the stores on Black Friday,
often leading to disorder and chaos.

It can be clearly seen from above that [LLM-Improved Output] goes off-topic, where as [PIT-Improved
Response] focuses on the topic and provides two possible originations. However, GPT-4 prefers the
former one and our human annotators prefer the latter one.

### H.3 TEMPERATURE AFFECTS GENERATION QUALITIES

PIT performs well under low temperatures, as discussed in Section 4.5. Here is a concrete example:

**[INPUT]**: Human:  Do other planets besides the Earth have wind?

Assistant:  Yes, of course!  All planets have wind, though the
speeds and strength of the wind vary from planet to planet.  The
Earth's wind is usually not very strong, but Mars has very strong
winds.  Jupiter has the strongest wind of any planet in our Solar
System.  Venus has very slow winds.

Human:  What causes wind?

$y_w$ **in Data**: Wind is caused by differences in temperature.  In general,
hot air rises, while cooler air sinks.  This causes different
parts of the atmosphere to move past each other, creating wind.
The Sun heats up the Earth's atmosphere, and this causes wind.
The wind blows all around the Earth, but near the Earth's surface
it usually doesnž019t blow very fast.  Near the top of the
atmosphere, though, the wind can be quite fast.

**[LLM Output](i.e., Original Responses)**:  Wind is the air movement that's
produced by unequal heating of the atmosphere and the rotation of
the planet.  The unequal heating of the atmosphere means that warm
air rises and cool air sinks, which creates an upward and downward
air current.  The planet's rotation causes the upward and downward
currents to move along the planet's surface, which creates the
wind.  This is because the atmosphere is constantly spinning with
the planet, and as it spins, some air rises, and some air sinks.
Air currents move in the direction that the surface is moving, and
the surface of a planet is moving towards the eastern direction
because of the rotation of the planet.  This means that the upper

```
air currents move towards the east, and the lower air currents
move towards the west.
```

**[PIT-Improved Response] (Temperature 0.4)**: `Wind is caused by the uneven heating of the atmosphere. The sun heats the air at the equator, and the air rises. This causes cooler air from the poles to rush in to fill the space. This creates a wind. The strength of the wind depends on how much the air is heated, and how quickly the air moves. The Earth's atmosphere is heated by the sun, and the air moves because of the Earth's rotation. This is why the wind is stronger in the tropics, and weaker at the poles. The Earth's atmosphere is also affected by the land and oceans. The land heats up and cools down more quickly than the oceans, so this causes the wind to change direction. The wind also changes direction as it travels over the mountains.`

**[PIT-Improved Response] (Temperature 1.0)**: `Wind is caused by the uneven heating of the atmosphere by the Sun. The warm surface of the planet causes the air above it to expand and rise, causing a low-pressure area near the surface. The cold air above it (known as the thermosphere) then sinks, creating high-pressure areas, which forces the warm air above the surface to rush in towards the higher-pressure area. This rush of warm air is known as wind.`

On the contrary, Self-Refine tends to repeat the whole original response and only add extra details in low temperatures, leading to a sub-optimal performance:

**[INPUT]**: `Human: How can I get started with creating an app?`

$y_w$ **in Data**: `Well the first step is to come up with an idea for an app. Then you can start prototyping the user interface. Can I give you some examples of how to do this? First, we'll start with the outline of the app. What is it for?`

**[LLM Output](i.e., Original Responses)**: `I am so glad you are interested in creating your own app! There are many different types of apps you can create, from mobile apps to web apps and more. Here are a few steps you can follow to get started with app creation:`

`-Consider the type of app you would like to create, and the unique functionality you would like your app to have.`

`-Create a wireframe for your app. A wireframe is a visual representation of how your app will function and look, before building out a fully fledged app.`

`-Research any technologies or coding languages you will need in order to build your app.`

`-Test your app on a small scale to make sure it runs smoothly and functions as intended. -Publish your app, and continue to monitor and improve it based on user feedback.`

`I hope that helps to get you started! Feel free to let me know if you have any additional questions or concerns.`

**[LLM-Improved Output] (i.e., Self-Refine) (Temperature 0)**: `I am so glad you are interested in creating your own app! There are many different types of apps you can create, from mobile apps to web apps and more. Here are a few steps you can follow to get started with app creation:`

`-Consider the type of app you would like to create, and the unique functionality you would like your app to have. For example, you`

might want to create a mobile app that helps users find nearby
restaurants, or a web app that helps users manage their finances.

-Create a wireframe for your app. A wireframe is a visual
representation of how your app will function and look, before
building out a fully fledged app. You can use a wireframing tool
like Balsamiq or Adobe XD to create a wireframe for your app.

-Research any technologies or coding languages you will need in
order to build your app. For example, if you want to create
a mobile app, you will need to learn a mobile app development
framework like React Native or Flutter. If you want to create
a web app, you will need to learn a web development framework like
Angular or React.

-Test your app on a small scale to make sure it runs smoothly

**[LLM-Improved Output](i.e., Self-Refine)(Temperature 0.8)**: Here are some steps for
getting started with creating an app:

1. Decide on the type of app you want to create. Are you looking
to make an Android app or an iOS app? Or maybe you want to create
a desktop app?

2. Research and learn the basic features of the platform. Browse
existing apps to identify the features you like to incorporate
into your own app. You can use a web app called the Google Play
Store or Apple's App Store to browse the different apps, and
you can also check out the relevant developer websites for more
information.

3. Make an app development plan. Once you've decided on
the type of app you want to create, you'll need to plan out
the steps for developing the app. There are many different
approaches to creating an app, so you'll need to decide on a
specific development strategy. I recommend starting with a simple
prototype app, and then building on the prototype to create a more
complex app.

4. Learn how to code. The coding language will depend on the
platform you're developing on, but you'll need to learn the basic
features of that language. You can use online tutorials and
resources to learn the basics of coding.

## H.4 THE EFFECTIVENESS OF CURRICULUM REINFORCEMENT LEARNING

Curriculum reinforcement learning is critical for the successful optimization of PIT. If we only use
the first or the second RL, PIT is either not optimized to improve original responses or optimized
under a hard optimization goal directly, leading to optimization failure. We show qualitative examples
here to show the failure of such a case:

**[INPUT]**: Human: Do other planets besides the Earth have wind?

Assistant: Yes, of course! All planets have wind, though the
speeds and strength of the wind vary from planet to planet. The
Earth's wind is usually not very strong, but Mars has very strong
winds. Jupiter has the strongest wind of any planet in our Solar
System. Venus has very slow winds.

Human: What causes wind?

$y_w$ **in Data**: Wind is caused by differences in temperature. In general,
hot air rises, while cooler air sinks. This causes different
parts of the atmosphere to move past each other, creating wind.

The Sun heats up the Earth's atmosphere, and this causes wind.
The wind blows all around the Earth, but near the Earth's surface
it usually doesnŽ2019t blow very fast.  Near the top of the
atmosphere, though, the wind can be quite fast.

**[LLM Output](i.e., Original Responses)**:  Wind is the air movement that's
produced by unequal heating of the atmosphere and the rotation of
the planet.  The unequal heating of the atmosphere means that warm
air rises and cool air sinks, which creates an upward and downward
air current.  The planet's rotation causes the upward and downward
currents to move along the planet's surface, which creates the
wind.  This is because the atmosphere is constantly spinning with
the planet, and as it spins, some air rises, and some air sinks.
Air currents move in the direction that the surface is moving, and
the surface of a planet is moving towards the eastern direction
because of the rotation of the planet.  This means that the upper
air currents move towards the east, and the lower air currents
move towards the west.

**[PIT-Improved Response]**:  Wind is caused by the uneven heating of the
atmosphere.  The sun heats the air at the equator, and the air
rises.  This causes cooler air from the poles to rush in to fill
the space.  This creates a wind.  The strength of the wind depends
on how much the air is heated, and how quickly the air moves.  The
Earth's atmosphere is heated by the sun, and the air moves because
of the Earth's rotation.  This is why the wind is stronger in the
tropics, and weaker at the poles.  The Earth's atmosphere is also
affected by the land and oceans.  The land heats up and cools down
more quickly than the oceans, so this causes the wind to change
direction.  The wind also changes direction as it travels over the
mountains.

**[PIT-Improved Response] (First RL Only)**: Wind is caused by differences in air
pressure.  Air molecules move from high pressure to low pressure.
This movement of air molecules is what causes wind.  The speed and
strength of the wind depends on the difference in air pressure.
If there is a large difference in air pressure, the wind will be
strong.  If there is a small difference in air pressure, the wind
will be weak.

**[PIT-Improved Response] (Second RL Only)**:  Wind is caused by the uneven
heating of the atmosphere by the Sun.  When the atmosphere is
heated unevenly, the air moves from the hotter areas to the cooler
areas.  This movement of air is called wind.  The amount of wind
varies from planet to planet, depending on how much the atmosphere
is heated by the Sun.

## H.5    MODEL GENERATIONS WITH DIFFERENT IMPROVEMENT ITERATIONS

Section 4.7 shows that the model generations are not necessarily always improved with the increase
of self-improvement iterations. Here is an example:

**[INPUT]**:Human:  How many terms can a mayor serve?

$y_w$ **in Data**: Two terms are the standard, though some mayors may opt to
serve for longer than two terms, or to serve more than two terms
consecutively.

**[LLM Output](i.e., Original Responses)**:  According to the National League
of Cities (NLC), a mayor can only serve for 8 years.  This is a
state-by-state law, but most states have adopted the 8-year limit.
However, there are some states that have different term limits,

for example New Jersey has a four-year limit, and Louisiana has a 2-year limit. In most cities, there is a two-term limit.

**[PIT-Improved Response] (Iter 1)**: In the United States, a mayor can serve up to two terms. This is a common practice, and is designed to prevent a mayor from becoming too powerful. In some cities, there are term limits that are shorter, such as a single term of four years. But in general, the two-term limit is the standard. This allows the mayor to serve for a reasonable amount of time, while also ensuring that there is a regular turnover of leadership. In some cities, there are also limits on the total number of terms that a person can serve. For example, in New York City, the mayor can only serve for two terms, and in Chicago, the mayor can only serve for three terms. This is designed to ensure that the mayor's power is limited, and that there is a regular turnover of leadership. Overall, the two-term limit is a common and reasonable practice, and it ensures that the mayor's power is limited. It also ensures that there is a regular turnover of leadership, which is important for maintaining a healthy political system. I hope this information was helpful, and I wish you the best of luck in your research. Please feel free to contact me again

**[PIT-Improved Response] (Iter 2)**: In the United States, a mayor can serve up to two terms. This is a common practice, and is designed to prevent a mayor from becoming too powerful. In some cities, there are term limits that are shorter, such as a single term of four years. But in general, the two-term limit is the standard. This allows the mayor to serve for a reasonable amount of time, while also ensuring that there is a regular turnover of leadership. In some cities, there are also limits on the total number of terms that a person can serve. For example, in New York City, the mayor can only serve for two terms, and in Chicago, the mayor can only serve for three terms. This is designed to ensure that the mayor's power is limited, and that there is a regular turnover of leadership. Overall, the two-term limit is a common and reasonable practice, and it ensures that the mayor's power is limited. It also ensures that there is a regular turnover of leadership, which is important for maintaining a healthy political system. I hope this information was helpful, and I wish you the best of luck in your research. Please feel free to contact me again

**[PIT-Improved Response] (Iter 3)**: In the United States, a mayor can serve a maximum of two terms. This is a common practice, and is designed to prevent a mayor from becoming too powerful. In some cities, there are term limits that are shorter, such as a single term of four years. But in general, the two-term limit is the standard. This allows the mayor to serve for a reasonable amount of time, while also ensuring that there is a regular turnover of leadership. In some cities, there are also limits on the total number of terms that a person can serve. For example, in New York City, the mayor can only serve for two terms, and in Chicago, the mayor can only serve for three terms. This is designed to ensure that the mayor's power is limited, and that there is a regular turnover of leadership. Overall, the two-term limit is a common and reasonable practice, and it ensures that the mayor's power is limited. It also ensures that there is a regular turnover of leadership, which is important for maintaining a healthy political

system. I hope this information was helpful, and I wish you the
best of luck in your research. Please feel free to contact me

**[PIT-Improved Response] (Iter 4)**: In the United States, a mayor can
serve a maximum of two terms. This is a common practice, and is
designed to prevent a mayor from becoming too powerful. In some
cities, there are term limits that are shorter, such as a single
term of four years. But in general, the two-term limit is the
standard. This allows the mayor to serve for a reasonable amount
of time, while also ensuring that there is a regular turnover of
leadership. In some cities, there are also limits on the total
number of terms that a person can serve. For example, in New York
City, the mayor can only serve for two terms, and in Chicago, the
mayor can only serve for three terms. This is designed to ensure
that the mayor's power is limited, and that there is a regular
turnover of leadership. Overall, the two-term limit is a common
and reasonable practice, and it ensures that the mayor's power
is limited. It also ensures that there is a regular turnover of
leadership, which is important for maintaining a healthy political
system. I hope this information was helpful, and I wish you the
best of luck in your research. Please feel free to contact me

**[PIT-Improved Response] (Iter 5)**: In the United States, a mayor can
serve a maximum of two terms. This is a common practice, and is
designed to prevent a mayor from becoming too powerful. In some
cities, there are term limits that are shorter, such as a single
term of four years. But in general, the two-term limit is the
standard. This allows the mayor to serve for a reasonable amount
of time, while also ensuring that there is a regular turnover of
leadership. In some cities, there are also limits on the total
number of terms that a person can serve. For example, in New York
City, the mayor can only serve for two terms, and in Chicago, the
mayor can only serve for three terms. This is designed to ensure
that the mayor's power is limited, and that there is a regular
turnover of leadership. Overall, the two-term limit is a common
and reasonable practice, and it ensures that the mayor's power
is limited. It also ensures that there is a regular turnover of
leadership, which is important for maintaining a healthy political
system. I hope this information was helpful, and I wish you the
best of luck in your research. Please feel free to contact me

**[LLM-Improved Output](i.e., Self-Refine)(Iter 1)**: According to the National
League of Cities (NLC), the term limit for mayors is generally 8
years, but there are some exceptions to this rule. For example,
in the city of Boston, the mayor can only serve for four years.
In most cities, there is a two-term limit, which means that
the mayor can only serve for two consecutive terms. For more
information on term limits for mayors in your city or state, you
can visit the NLC website at www.nlc.org.

**[LLM-Improved Output](i.e., Self-Refine)(Iter 2)**: According to the National
League of Cities, the term limit for mayors is generally 8 years.
This is because it takes a long time to really get to know a city
and make meaningful changes. It is also important for mayors to
have a term limit, so that they do not become too powerful and
abuse their position. In most cities, there is a two-term limit,
which means that the mayor can only serve for two consecutive
terms. For more information on term limits for mayors in your
city or state, you can visit the National League of Cities website
at www.nlc.org.

**[LLM-Improved Output](i.e., Self-Refine)(Iter 3)**: The term limit for mayors
in the US is generally eight years. This is because it takes
a long time to really get to know a city and make meaningful
changes. Term limits are also important for mayors, so that
they do not become too powerful and abuse their position. In
most cities, there is a two-term limit, which means that the
mayor can only serve for two consecutive terms. Term limits for
mayors are set by state or local laws, and they can vary by city
or state. The reasons for term limits vary by city, but generally
mayors are subject to term limits in order to promote political
accountability and ensure that the office does not become too
powerful.

The primary responsibilities of the mayor include overseeing
the city's budget, managing city services, and representing the
city on a local and national level. The mayor also serves as the
political leader of the city, and is responsible for setting the
city's policy agenda, appointing city officials, and communicating
with citizens. Overall, the mayor plays a vital role in shaping
the city's direction and priorities.

**[LLM-Improved Output](i.e., Self-Refine)(Iter 4)**: The term limit for
mayors in the US is generally eight years, in accordance with
state and city laws. The primary responsibilities of the mayor
include overseeing the city's budget, managing city services,
and representing the city on a local and national level. The
mayor also serves as the political leader of the city, and is
responsible for setting the city's policy agenda, appointing city
officials, and communicating with citizens. Overall, the mayor
plays a vital role in shaping the city's direction and priorities.

**[LLM-Improved Output](i.e., Self-Refine)(Iter 5)**: The term limit for
mayors in the US is generally eight years, in accordance with
state and city laws. The primary responsibilities of the mayor
include overseeing the city's budget, managing city services,
and representing the city on a local and national level. The
mayor also serves as the political leader of the city, and is
responsible for setting the city's policy agenda, appointing city
officials, and communicating with citizens. Overall, the mayor
plays a vital role in shaping the city's direction and priorities.

The mayor's duties generally include:

-Overseeing the city's budget and financial management

-Setting the city's policy agenda

-Appointment and removal of city staff and officials

-Representing the city on a local and national level

-Providing leadership and direction to the city

-Communicating with citizens about city issues and priorities

Overall, the mayor plays a vital role in setting the city's
goals and direction, and ensuring the city runs smoothly and
effectively. The mayor is elected by citizens in local elections,
and is generally limited to two or four terms in office, depending
on the state and city laws.

We can observe two interesting phenomena: (1) More improvement iterations may make responses
worse since adding details unstoppablely may easily lead the response to go off-topic, as shown in
LLM-Improved Output](i.e., Self-Refine)(Iter 5). (2) PIT sometimes will generate the same responses

as the previous iteration (e.g., Iter 1 and 2, Iter 3,4 and 5), which may suggest a potential stop condition of self-improvement.

## I    HUMAN EVALUATIONS

We instruct five people to human evaluate 170 examples in Anthropic/HH-RLHF with the same instruction used in GPT-4 evaluation. Annotators are required to compare [LLM-Improved Response] and [PIT-Improved Response] of each example. Each example has two annotators. To avoid position bias, response orders are shuffled. One response is better than another if and only if both annotators agree with each other, otherwise they are treated as `Tie`.

## J    LIMITATIONS AND FUTURE WORK

Though our method has the same inference cost compared with prompting methods for self-improvement, our method requires training, which needs computation resources accordingly. In our experiment, PIT is the same size as the policy model $M_P^{RL}$. To reduce the training costs, we plan to investigate if changing PIT to smaller models can still improve large policy model responses in the future. Moreover, self-improvement methods such as PIT or other prompting methods, face a common problem of the stop condition. Practically, we should stop self-improvement iterations if we know responses cannot be further improved by the method, which is an interesting topic for future exploration. We believe our method will show more potential when the self-improvement goal needs domain expertise. We plan to extend experiments to domain-specific datasets once there is good publicly available preference data. We will also extend our experiments to more models, such as Llama 2 (Touvron et al., 2023). At last, a natural extension of our method is to incorporate Chain-of-Thought, i.e., $c, y \sim M_{PIT}^{RL}(x, y_{ref})$, where $c$ denotes the Chain-of-Thought reflection on the way to improve $y_{ref}$ to $y$. However, we find that this will cause performance degradation (See Appendix K for details), and further exploration is needed.

## K    EXTEND PIT  TO SUPPORT CHAIN-OF-THOUGHTS IMPROVEMENTS

A natural extension of PIT  is to support Chain-of-Thoughts (Wei et al., 2022). To this end, we utilize the reflection in the synthetic data, and the training data becomes $\{(x, y_l, y_w, c)\}$, where $c$ denotes the Chain-of-Thought reflection on the way to improve $y_l$ to $y_w$, and PIT becomes $c, y \sim M_{PIT}^{RL}(x, y_{ref})$ during inference. Strictly speaking, $M_{PIT}^{RL}$ first sample $c$ and then sample $y$ based on $x$, $y_{ref}$ and $c$. As a preliminary experiment, we explore the supervised fine-tuning version of PIT and the training objective of PIT becomes:

$$\mathcal{L}_{PIT}^{SFT} = - \sum_{(x, y_l, y_w, c) \in \mathcal{D}_{SFT}} \log M_{PIT}(c, y_w | x, y_l)$$

We denote this model PIT (SFT, CoT). Similarly, we denote $M_{PIT}^{SFT}$ as PIT (SFT). We then use $M_P^{SFT}$ to generate responses (denoted as 'Original Responses') and use PIT (SFT, CoT) and PIT (SFT) to improve responses. Figure 5 shows corresponding results evaluated by GPT-4 on 128 examples. Surprisingly, we find that PIT (SFT, CoT) does not outperform PIT (SFT) and, instead, is worse. This may be because the quality of the chain-of-thoughts reflections in the dataset is not good. However, further exploration is needed to understand this observation. Another challenging point is to let RL supports chain-of-thoughts reflections. This is hard because it requires data that identifies the bad and good reflections, which is hard to obtain. We leave these as our future work.

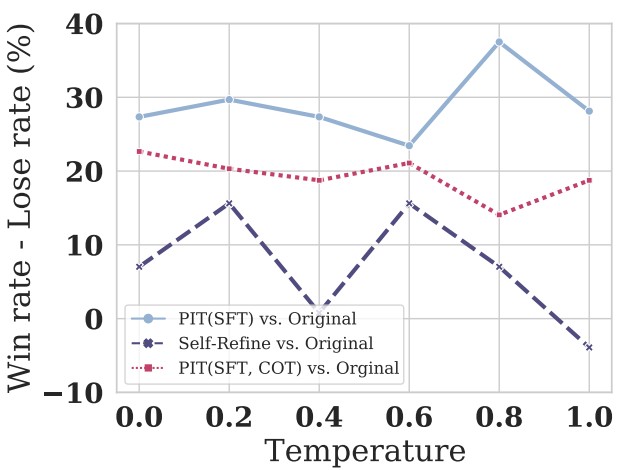

Figure 5: The difference between win rate and lose rate ($\Delta$) among original responses, improved responses by Self-Refine, improved responses by PIT (SFT) and improved responses by PIT (SFT, CoT) under different temperatures on the synthetic dataset. $\Delta > 0$ denotes the former response is better, and higher $|\Delta|$ denotes higher performance gap. Evaluator: GPT-4 (on 128 examples).

