# OpenReview forum: "Enabling Lanuguage Models to Implicitly Learn Self-Improvement"
_ICLR.cc/2024/Conference — ICLR 2024 poster_

### Official Review · Reviewer_LeAH · 2023-10-29

**Soundness:** 3 good
**Presentation:** 2 fair
**Contribution:** 3 good
**Rating:** 6
**Confidence:** 3

**Summary:**

The paper proposes a framework, PIT, that trains an additional model that improves the output of an LLM with RLHF. The reward is set to be the preference gap between the improved output and the original LLM output, i.e. implicitly trained with the improvement goal of better aligning with human preferences.

**Strengths:**

1. The paper introduces an interesting framework, PIT, that can learn how to improve the LLM output in a RLHF way, which does not require a significant amount of human annotation and prompt design.
2. Instead of updating the original LLM as the original RLHF, PIT proposes to update the model that targets improving the output for the original LLM, which is a novel technique.
3. The paper evaluates the PIT technique with the previous self-refine method that also tries to improve the LLM output, and demonstrates a better result. It also shows a set of ablation studies that show the effectiveness of each component.

**Weaknesses:**

1. The paper motivates why using RLHF in output improvement is better than just prompting the LLM to give feedback, but it is unclear why we should apply RLHF for the improvement, instead of directly applying RLHF to the original LLM. It would be better to see the comparison in the evaluation.
2. Comparing PIT with Self-Refine seems unfair as PIT requires more human annotation as it requires additional human preference labels.
3. The description of PIT is a bit unclear as the model used in each component of the method is not clearly stated.

**Questions:**

NA

---

> ### Author Response · Authors · 2023-11-13
> **Response to reviewer LeAH**
>
> Thank you for the thoughtful feedback! We are happy that you find our approach interesting, novel, and effective! We understand you may have some confusion and concerns, which we address as follows.
>
> > ### **W1: The paper motivates why using RLHF in output improvement is better than just prompting the LLM to give feedback, but it is unclear why we should apply RLHF for the improvement, instead of directly applying RLHF to the original LLM. It would be better to see the comparison in the evaluation.**
>
> The original LLM (policy model in Section 3 and “LLMs” in Figure 1) that generates the initial response is already RLHF-finetuned. PIT takes initial responses generated by the original LLM as input, and generates improved responses. PIT and the original LLM were finetuned with the same starting pre-trained checkpoint and data, but PIT is not a continual training of the original LLM (Continual training may not be suitable since their input formats are different. Appendix A illustrates more details). PIT only utilizes the output of the original LLM during training and inference time and does not use its weights. We compare PIT with (1) original responses given by the original LLM and (2) Improved responses given by Self-Refine on the original LLM to show PIT is better than the original LLM with Self-Refine.
>
> We apologize for not making ourselves clearer and causing you some potential confusion. We demonstrate our workflow in Appendix A in the revised paper to make our approach clearer and more organized. We hope these extra demonstrations are helpful for a better understanding of our approach and can clarify this concern.
>
> > ### **W2: Comparing PIT with Self-Refine seems unfair as PIT requires more human annotation as it requires additional human preference labels.**
>
> The training data of PIT is exactly the same as the data we use to train the RLHF-finetuned model. Thus, PIT needs **no extra data**. A more detailed workflow demonstration can be found in Appendix A in the revised paper. Again, we apologize for the confusion and for not making this point clearer.
>
> > ### **W3: The description of PIT is a bit unclear as the model used in each component of the method is not clearly stated.**
>
> We make a more detailed workflow demonstration containing the input format training algorithm blocks in Appendix A in the revised paper. We believe general response GR2 will also be helpful for understanding. We hope now the framework is clearer.
>
>
> We hope our reply addresses your questions, and please let us know if you have further questions. We would be happy to continue the discussion if needed.

---

> > ### Author Response · Authors · 2023-11-21
> >
> > Dear reviewer LeAH,
> >
> > We would like to kindly remind you that the rebuttal period is going to end soon, and we hope to hear your valuable feedback. Thank you again for putting your time into reviewing our paper :)

---

> > ### Comment · Reviewer_LeAH · 2023-11-23
> > **RE: Response to reviewer LeAH**
> >
> > Thanks to the authors for the responses! They solved most of my concerns. Thus, I increased the score to 6.

---

> > > ### Author Response · Authors · 2023-11-23
> > >
> > > Thank you for your time in reviewing our paper! We appreciate your feedback, which makes the paper in a better form.

---

### Official Review · Reviewer_tfvt · 2023-10-30

**Soundness:** 3 good
**Presentation:** 4 excellent
**Contribution:** 3 good
**Rating:** 6
**Confidence:** 3

**Summary:**

This paper proposes an RLHF-style fine-tuning routine to allow large language models (LLMs) to self-improve. The authors propose PIT that modifies the RLHF routine in all three stages: at the supervised fine-tuning stage, PIT maximises the likelihood of the better response conditional on both the instruction and the worse outputs (instead of the instructions only). In the reward model training stage, PIT encodes 4 pairwise relations between good and worse outputs instead of simply optimising the reward gap between better and worse responses. In the RL stage, instead of defining reward only in terms of supervised finetuned models and the RL model, PIT utilises multiple stages of RL to improve both over the annotated examples and iteratively improve over its previous responses (thereby achieving self-improvement). Experiments are done on 3 RLHF datasets, and the authors show that their proposed methods compare favourably both over the original response and over self-refine, a prompt-based self-improvement method.

**Strengths:**

- I found the paper’s motivation of implicitly self-improving from preference data instead of explicit rubrics construction to be well thought out, convincing and likely of interest to the community.
- The execution is largely reasonable and intuitive and I like the pairwise relation encoding in the methodology, although there is room for improvement in this area; see weaknesses. The idea of curriculum RL to iteratively and continuously improve on LLM outputs is also intriguing.
- The experimental section is largely thorough, and the improvement over baseline and prompting-based self-refine seems largely convincing. Experimental support is also provided for some (but not all) of the critical designs, such as the use of two (or more) stages of reinforcement learning and the indispensability of each stage.
- The paper is generally well-written and clear.

**Weaknesses:**

- The computational cost and the execution difficulty should be more clearly stated: while the method seems to lead to a stronger gain than self-refine, such an improvement is not always significant and sometimes self-refine seems to be stronger by some metric (e.g., in terms of GPT-4 evaluation), although the authors have given possible explanations as to why they occur. On the other hand, self-refine as a prompting-based method is much easier to execute and cheaper I think the paper would benefit if the authors would give more detailed account on the computational cost, including a comparison with the baseline methods.
- Some claims are qualitatively argued rather than empirically validated. An example is the use of pairwise relation training in Eq (2), a key component of the algorithm’s design. The authors largely provide intuitive explanations against the simpler alternative in favour of the more complicated design the paper adopted, but no empirical validation is provided.

## Minor
- Typo in openreview title: Lanuguage -> Language
- First line beneath caption of Fig 1: rubics -> rubrics

**Questions:**

- Address my concerns in weaknesses.
- It seems to me that PIT is run on top of an RLHF-finetuned model (i.e., the model generating “original” outputs in experiments)? if so, this point should be more clearly stated. If not, why not? In that case, RLHF should be an obvious baseline to compare against given that PIT is closely formulated based on the original RLHF (I gave the benefit of the doubt in the preliminary review on this point, pending clarification in rebuttal).
- Do you observe that the reward model remains discriminative after multiple rounds of improvements? It seems to me that the reward model is trained on the original y_w and y_l annotated responses only, but the new outputs should be even better than y_w after a few iterations.

-- **Post-rebuttal** --

I thank the authors for the detailed feedback, which largely addressed my concerns. I also read other reviews, and I will stick to my rating that recommends acceptance. I think the discussion regarding prompting methods might be better qualified, though, as even if the proposed method uses fewer tokens, prompting-based methods have the strength that they do not require model weight adjustments, and only forward passes are required. As discussed in the original review, the results compared to baselines can be occasionally somewhat mixed, so I am unable to give an even higher score. Nonetheless, I believe this paper is of value to the community and can be accepted at ICLR.

---

> ### Author Response · Authors · 2023-11-13
> **Response to reviewer tfvt (1/3)**
>
> We thank the reviewer for the detailed and insightful feedback! We are excited that you think our method is convincing, interesting, reasonable, intuitive, and intriguing! Here are our responses to clarify your questions:
>
> > ### **W1.1: The computational cost and the execution difficulty should be more clearly stated: while the method seems to lead to a stronger gain than self-refine, such an improvement is not always significant and sometimes self-refine seems to be stronger by some metric (e.g., in terms of GPT-4 evaluation), although the authors have given possible explanations as to why they occur.**
>
> We understand the reviewer’s concern about GPT-4 metrics, so we introduce human evaluation in this case. GPT-4 is well-known for many biases, such as position bias [1,2], format bias [3], and knowledge bias [3]. Therefore, we introduce human evaluation when GPT-4 disagrees with the reward model. Our human evaluation detects several false negative cases in GPT-4, like long-content preference, as stated in Section 4.4. Therefore, we trust human evaluation in this case, which shows PIT is still apparently better than Self-Refine (Table 1).
>
> [1] Wang, Peiyi, et al. "Large language models are not fair evaluators." arXiv preprint arXiv:2305.17926 (2023).
>
> [2] Zheng, Lianmin, et al. "Judging LLM-as-a-judge with MT-Bench and Chatbot Arena." arXiv preprint arXiv:2306.05685 (2023).
>
> [3]Zhu, Lianghui, Xinggang Wang, and Xinlong Wang. "JudgeLM: Fine-tuned Large Language Models are Scalable Judges." arXiv preprint arXiv:2310.17631 (2023).
>
>
> > ### **W1.2 On the other hand, self-refine as a prompting-based method is much easier to execute and cheaper I think the paper would benefit if the authors would give more detailed account on the computational cost, including a comparison with the baseline methods**
>
> Thanks for pointing out this important discussion. A short answer to the computational cost concern is that our method does not bring extra computational cost compared to Self-Refine during inference and, in principle, will even have a bit lower computational cost due to fewer token inputs (without rubrics). A more detailed explanation can be found in the general response GR2.
>
> For the execution difficulty, we would like to address that Self-Refine is more challenging to execute than it looks since it needs careful prompt/rubric design, which is one of our motivations. From the end-user perspective, our method is more friendly since the end-user only needs to provide the query and a candidate response without putting effort into designing prompts.
>
> Overall, we appreciate the reviewer pointing out this important discussion, and we already added this to Appendix B of the revised paper.

---

> > ### Author Response · Authors · 2023-11-13
> > **Response to reviewer tfvt (2/3)**
> >
> > > ### **W2: Some claims are qualitatively argued rather than empirically validated. An example is the use of pairwise relation training in Eq (2), a key component of the algorithm’s design. The authors largely provide intuitive explanations against the simpler alternative in favour of the more complicated design the paper adopted, but no empirical validation is provided.**
> >
> > Thanks for the constructive feedback. We made many efforts to rule out alternative options to our proposed Eq (2). Since it is not applicable to try all possible solutions, we rule out other methods based on prior work findings and our own pilot experiments. Here is our detailed selection process:
> >
> > 1. **Do we need to train a separate reward model to model reward gaps?**
> >
> > The first question in developing the reward model is if we can directly use $R\_p$ (i.e., the vanilla reward model that takes $x$ and $y$ as inputs and returns a scalar reward $r$) to compute the reward gap between two responses $y\_1$ and $y\_2$, or we need to develop another reward model that directly calculates the reward gap. The most straightforward implementation using $R\_p$ is to compute the subtraction of individual rewards of $y\_1$ and $y\_2$ obtained by $R\_p$. However, previous works show that $R_p$ fails to faithfully reflect the response quality when $r$ is high [4, Figure 25]. That is to say, if $r\_1$ and $r\_2$ are higher than a threshold, then $r\_1 < r\_2$ does not necessarily denote $y\_1$ is worse. This phenomenon is possible because of the poor calibration of the reward model [5].  Moreover, other work [6] shows that directly modeling the reward gap brings less noise and performs better than computing the subtraction of individual rewards. Therefore, we decided to train a reward model that directly models reward gaps. (This part is also briefly mentioned in Section 3.3).
> >
> > 2. **What training objectives are suitable to grasp the reward gap?**
> >
> > Next, we need to decide the training objective. In the beginning, the training objective is simply maximizing the reward gap $r\_{\text{gap}}(x, y\_w, y\_l)$ and minimizing $r\_{\text{gap}}(x, y\_l, y\_w)$, which is similar to Equation (2) but without $r\_{\text{gap}}(x, y\_w, y\_w)$ and $r\_{\text{gap}}(x, y\_l, y\_l)$. However, we observe that LLM finds a shortcut that ignores the third argument, thus degenerating to $R_p$ and failing to grasp the reward gap. Therefore, we add $r\_{\text{gap}}(x, y\_w, y\_w)$ and $r\_{\text{gap}}(x, y\_l, y\_l)$ to prevent the degeneration.
> >
> > We also explore the possibility of using pointwise rather than pairwise training signals used above. For example, we tried to explicitly assign $r\_{\text{gap}}(x, y\_w, y\_l) = 1$, $r\_{\text{gap}}(x, y\_l, y\_w) = 0$ and $r\_{\text{gap}}(x, y\_l, y\_l) = r\_{\text{gap}}(x, y\_w, y\_w) = 0.5$ and use MSE to train the reward model. We find this implementation is less effective than our proposed approach (e.g., Equation (2)) on the held-out reward model test set of 500 examples (~5% accuracy drop). We think this phenomenon is because the inductive bias of assigning rewards to 1, 0.5, and 0 is not preferred in the data distribution.
> >
> > Based on the above analysis, Equation (2) is chosen as our final choice. We thank you again for raising this interesting discussion, and we added this discussion to our revised paper Appendix C.
> >
> >
> >
> > [4] Bai, Yuntao, et al. "Training a helpful and harmless assistant with reinforcement learning from human feedback." arXiv preprint arXiv:2204.05862 (2022).
> >
> > [5] Peng, Baolin, et al. "Stabilizing RLHF through Advantage Model and Selective Rehearsal." arXiv preprint arXiv:2309.10202 (2023).
> >
> > [6] Zhao, Yao, et al. "Slic-hf: Sequence likelihood calibration with human feedback." arXiv preprint arXiv:2305.10425 (2023).

---

> > > ### Author Response · Authors · 2023-11-13
> > > **Response to reviewer tfvt (3/3)**
> > >
> > > > ### **Minor: Typo**
> > >
> > > Thanks for your careful feedback. We’ve fixed this in the revised version.
> > >
> > > > ### **Q2: It seems to me that PIT is run on top of an RLHF-finetuned model (i.e., the model generating “original” outputs in experiments)? if so, this point should be more clearly stated. If not, why not? In that case, RLHF should be an obvious baseline to compare against given that PIT is closely formulated based on the original RLHF (I gave the benefit of the doubt in the preliminary review on this point, pending clarification in rebuttal).**
> > >
> > > PIT and the RLHF-finetuned policy model (i.e., “LLMs” in Figure 1) were finetuned with the same starting pre-trained checkpoint and data, but PIT is not a continual training of RLHF-finetuned model (continual training may not be suitable since their input formats are different. Appendix A illustrates more details). PIT only utilizes the output of the RLHF-finetuned model during training and inference time, and does not update the weights of the RLHF-finetuned model. PIT takes initial responses generated by the RLHF-finetuned model as input, and generates improved responses. Therefore, in our experiment design, we compare PIT with (1) original responses given by the RLHF-finetuned model and (2) Improved responses given by Self-Refine on the RLHF-finetuned model to show PIT is better than the RLHF-finetuned model with Self-Refine.
> > >
> > >
> > > > ### **Q3: Do you observe that the reward model remains discriminative after multiple rounds of improvements? It seems to me that the reward model is trained on the original $y\_w$ and $y\_l$ annotated responses only, but the new outputs should be even better than $y\_w$ after a few iterations.**
> > >
> > > We believe that the reward model is still discriminative in all three datasets we use. During training, we observed the reward model is quite good at generalization in both RLHF finetuning and PIT training. For example, we find an RLHF-finetuned model (i.e., not PIT, but the model to generate “original” outputs) will tend to generate much longer and more detailed responses compared to its SFT counterpart, though $y\_w$ and $y\_l$ are much shorter in the SFT and reward model training dataset. During the PIT training, the reward model still generalized well on better new outputs (In equation 5). We acknowledge it will be interesting to see the generalization boundary of the reward model in both RLHF fine-tuning and PIT training. However, it is out of this paper’s scope.
> > >
> > >
> > > We hope we make a satisfactory reply to address your questions and concerns. Please let us know if you have further questions, and we will be happy to continue the discussion.

---

> > > > ### Author Response · Authors · 2023-11-21
> > > >
> > > > Dear reviewer tfvt,
> > > >
> > > > We would like to kindly remind you that the rebuttal period is going to end soon, and we hope to hear your valuable feedback. Thank you again for putting your time into reviewing our paper :)

---

> ### Author Response · Authors · 2023-11-23
> **Response to reviewer's post-rebuttal comment**
>
> We thank the reviewer for the valuable feedback. We sincerely appreciate your time in reviewing our paper.
>
> A potential method to tackle the training computational cost issue is to incorporate PIT into the original model by jointly training the original model on both the PIT training objective and RLHF objective, so that we can integrate PIT functionality into the original model without training a separate model. We thank the reviewer for bringing this discussion; this could be an interesting future work :)

---

### Official Review · Reviewer_c4bA · 2023-11-01

**Soundness:** 3 good
**Presentation:** 3 good
**Contribution:** 3 good
**Rating:** 6
**Confidence:** 3

**Summary:**

This paper proposes a novel framework, implicit self-improvement, to learn from human preference data. Instead of optimizing the response quality, their method maximizes the gap of the pair of responses. The experiments on three datasets demonstrate the effectiveness of their method.

**Strengths:**

1. The paper proposes a novel method to implicitly self-improve from data. In this way, PIT can iteratively improve responses by repeating the self-improvement process.
2. The authors find a practical way to implement their ideas for maximizing the gaps between responses for implicit self-improvement. They conduct experiments and analyses to verify PIT's effectiveness.

**Weaknesses:**

1. In the experiment, the authors compare PIT with prompt-based methods (self-refine), while there is a lack of comparison with other reinforcement learning related methods like [1]. Can other RL methods help to self-improve the response?
[1] Song F, Yu B, Li M, et al. Preference ranking optimization for human alignment[J]. arXiv preprint arXiv:2306.17492, 2023.

**Questions:**

None.

---

> ### Author Response · Authors · 2023-11-13
> **Response to reviewer c4bA**
>
> We appreciate the reviewer for the helpful feedback. We are happy to see that you find our method novel and practical! Here are our responses to your concerns:
>
> > ### **W1: In the experiment, the authors compare PIT with prompt-based methods (self-refine), while there is a lack of comparison with other reinforcement learning related methods like [1]. Can other RL methods help to self-improve the response? [1] Song F, Yu B, Li M, et al. Preference ranking optimization for human alignment[J]. arXiv preprint arXiv:2306.17492, 2023.**
>
>
> We thank the reviewer for pointing out related resources. We added a discussion comparing our algorithm with the mentioned PRO paper in Section 3 (highlighted with green color). In short: PRO and similar papers such as DPO [1], focus on LLM alignment which is not what our work aims for. We focus on triggering the self-improvement capability implicitly from data. Therefore, PIT is not directly comparable to these methods.
>
> More specifically, RLHF, DPO, and PRO focus on “How to make LLM generate responses preferred by humans given an input $x$”, whereas PIT focuses on “How to get a better response preferred by humans given an input $x$ and a reference response $y$, where the reference response is generated by policy models (LLMs) trained with RLHF, DPO or PRO.” Appendix A shows more details of PIT, which may help understand that PIT has different optimization goals compared to PRO, etc.
>
> [1] Rafailov, Rafael, et al. "Direct preference optimization: Your language model is secretly a reward model." arXiv preprint arXiv:2305.18290 (2023).
>
> We hope our reply addresses your concerns, and please let us know if you need further clarification. We are happy to continue discussions if needed!

---

> > ### Author Response · Authors · 2023-11-21
> >
> > Dear reviewer c4bA,
> >
> > We would like to kindly remind you that the rebuttal period is going to end soon, and we hope to hear your valuable feedback. Thank you again for putting your time into reviewing our paper :)

---

### Official Review · Reviewer_xn4T · 2023-11-03

**Soundness:** 4 excellent
**Presentation:** 3 good
**Contribution:** 3 good
**Rating:** 6
**Confidence:** 3

**Summary:**

The paper introduces the PIT framework, an innovation aimed at enhancing the quality of text generation in LLMs. PIT offers a novel twist on RLHF by reformulating its objectives towards increasing the gap in quality between a generated response and a original reference response. This work bypasses the need for costly and complex explicit rubrics. The paper demonstrates the superiority of PIT over conventional self-refinement methods across various datasets.

**Strengths:**

The paper's main contribution lies in its innovative approach to reformulating the objectives of RLHF, particularly within the context of supervised fine-tuning reward model training, and reinforcement learning. It is particularly notable for introducing a novel reward model that prioritizes the quality gap between two responses from LLMs. This unique angle encourages the model to continually refine its output by comparing it to a reference response. The method has the potential to shift the paradigm in how LLMs are self-improved for better alignment with human preferences without the designing rubrics or prompts.

**Weaknesses:**

1. The framework may not incorporate new information or data during the self-improvement cycle, potentially limiting the scope of learning to alignment with initial human preferences.
1. The iterative nature of the proposed self-improvement process could result in increased computational time and cost in inference.

**Questions:**

1. Can you clarify if the three models depicted in Figure 1, i.e. LLMs, PIT, and PIT Reward Model? Are they distinct or the same model?
1. How to train with objectives (4) and (5)? An algorithm block is easy to illustrate.
1. The paper lacks a detailed algorithmic flow for training with objectives (4) and (5). Could you provide an algorithmic block to illustrate the training process?
1. How to determine the number of iterations for the self-improvement process for inference?

---

> ### Author Response · Authors · 2023-11-13
> **Response to reviewer xn4T**
>
> We thank the reviewer for the constructive feedback! We are glad you find our method novel and the potential to guide future directions. We address your concerns in the following paragraphs.
>
> > ### **W1: The framework may not incorporate new information or data during the self-improvement cycle, potentially limiting the scope of learning to alignment with initial human preferences.**
>
> We understand the reviewer's concern about the scenario of new incoming preferences. This is a challenging question for the conventional RLHF, too, not only our approach. In the conventional RLHF setting, one needs to re-train the reward model or ensemble different reward models. PIT can adopt the same technique to handle new incoming preferences as conventional RLHF.
>
>
> >### **W2: The iterative nature of the proposed self-improvement process could result in increased computational time and cost in inference.**
>
> In fact, the inference cost is the same as Self-Refine and we apologize for not directly pointing this out in the paper. Detailed discussions can be found in the general response GR2.
>
> Both Self-Refine and PIT iteratively replace candidate responses with improved responses (See Algorithm 2 in Appendix A in the revised version for more details). We evaluate the self-improvement of both methods under one iteration (Table 1, 2, and Figure 3) and multiple iterations (Table 3). Both methods have identical computational costs given the same number of iterations.
>
>
> >### **Q1: Can you clarify if the three models depicted in Figure 1, i.e. LLMs, PIT, and PIT Reward Model? Are they distinct or the same model?**
>
> We apologize for the confusion. In our experiments, all models start with the same pre-trained language model, i.e., PaLM 2 (Bison). “LLM” (or the policy model $\text{M}\_\text{P}^\text{RL}$ in Section 3) is the model fine-tuned on the dataset (e.g., HH-RLHF) following the normal RLHF pipeline. “PIT” (i.e., $\text{M}\_\text{PIT}^\text{RL}$ in Section 3) is the model fine-tuned with our reformulated RLHF pipeline (Equation 4 and 5) on the same dataset (e.g. HH-RLHF). “PIT Reward Model” (i.e., $\text{R}\_\text{PIT}$ in Section 3) is the model fine-tuned with our reformulated reward model training objective (equation 2) on the same dataset (e.g., HH-RLHF). “PIT reward model” offers the reward signal in equations 4 and 5 when training “PIT”.
>
> During inference, we first get a candidate response from “LLM”, then self-refine will design prompts and let “LLM” self-improve the candidate response. Differently, our approach uses “PIT” to improve the candidate response.
>
> A more formal and detailed demonstration including input formats, training, and inference details can be found in Appendix A in the revised paper. We also added some clarifications in Section 3 to help readers understand their differences (modifications are highlighted with green color).
>
>
> >### **Q2 and Q3: How to train with objectives (4) and (5)? An algorithm block is easy to illustrate. The paper lacks a detailed algorithmic flow for training with objectives (4) and (5). Could you provide an algorithmic block to illustrate the training process?**
>
> We have provided these details in our revised paper Appendix A. In short, to train with objective (4), we sample $x$ and $y\_\text{ref}$ from data $\mathcal{D}\_\text{RL}$ and use RL for optimization. To train with objective (5), we sample $x$ from data $\mathcal{D}\_\text{RL}$ and $y\_\text{ref}$ from $\text{M}\_\text{P}^\text{RL}(\cdot | x)$, and then use RL for optimization.
>
>
> >### **Q4: How to determine the number of iterations for the self-improvement process for inference?**
>
> This is a great question! It is a common problem for all self-improvement methods during inference. It is challenging to determine the stopping criteria for both Self-Refine and PIT. We show preliminary analysis in the paper (Section 4.7) -- there is no guarantee that every iteration can lead to improvement. Table 3 shows that more iterations do not denote better response qualities for both Self-Refine and our method. Nevertheless, our method is better than Self-Refine under all iterations, showing its effectiveness.Further research is needed to investigate the stopping criteria of self-improvement methods. One potential method is gradually reducing the temperature for the next iteration until the response is repeated.
>
>
>
>
> We hope we clarify your concerns and questions, please let us know if you have further questions and we will be happy to discuss further.

---

> > ### Author Response · Authors · 2023-11-21
> >
> > Dear reviewer xn4T,
> >
> > We would like to kindly remind you that the rebuttal period is going to end soon, and we hope to hear your valuable feedback. Thank you again for putting your time into reviewing our paper :)

---

### Author Response · Authors · 2023-11-13
**General Response**

We thank all reviewers for their valuable time in reviewing our paper! We are happy that reviewers find our method novel (all reviewers), effective (Reviewer LeAH), practical (Reviewer c4bA), interesting (Reviewer LeAH, tfvt), convincing (Reviewer tfvt) and has potential benefits for future research directions (Reviewer tfvt, xn4T). We address the common concerns in this general response, mainly to better demonstrate our approach.

> ### **GR1: The working flow**

Reviewers point out that an algorithm block will help understand the whole framework. Therefore, we added an extra Appendix A in our revised paper, which includes input formats, training algorithms, and inference algorithms. We hope these extra demonstrations help readers understand our approach better.


> ### **GR2: Inference and computational cost.**

Some reviewers also raise questions about the inference time efficiency. We hope to address that our approach has the same (or a bit lower) computational overhead compared with Self-Refine. The inference process is shown in Algorithm 2 in Appendix A of the revised paper. Here is a detailed explanation:

The inference prompt of Self-Refine has the following format:

> Human: Given a prompt {x} and a candidate response {$y\_\text{ref}$} provided by one assistant, you need to improve the response to be more helpful. Helpful means {rubric}. Assistant:

PIT uses a much simpler prompt with only special tokens as splitters:

> Human: {x} Assistant: <<Candidate>> {$y\_\text{ref}$} <<Improved>>

The {} denotes the concrete content, and <<>> denotes the special token. Both methods will return an improved response. The difference is that PIT does not need to explicitly write prompts since self-improvement is learned during our proposed training framework. Both methods use a one-time inference to get an improved response, and PIT, in principle, has **the same computational cost** as Self-Refine during inference,  or strictly speaking, even fewer computational costs due to the fewer input tokens (without complex rubrics).

We’ve included this discussion in Appendix B of the revised paper.


> ### **GR3: Paper revision.**

According to the reviewers' suggestions, we detail our framework in Appendix A, add discussions about the computational cost in Appendix B, and add discussions about the rationality of Equation 2 in Appendix C. We also insert pointers and extra explanations in the main body (Section 3) to guide readers to read the corresponding discussions and help readers understand the approach. The modifications are highlighted in green texts.

---

### Meta-Review · Area_Chair_8JhD · 2023-12-21

**Metareview:**

The paper proposes a method PIT that trains an additional model that improves the output of an LLM with RLHF. The reward is set to be the preference gap between the improved output and the original LLM output, i.e. implicitly trained with the improvement goal of better aligning with human preferences.

All reviewers acknowledge the novel approach based on RLHF proposed in this paper and experimental justifications.
The authors may want to add additional comparison with prompt-based self-refine/self-improve/learning-based approaches, and discussion on computational cost.

**Justification For Why Not Higher Score:**

The comparison and discussion against prompt-based approaches are not clear.

**Justification For Why Not Lower Score:**

The idea and method exhibit adequate novelty.

---

### Decision · Program_Chairs · 2024-01-16

Accept (poster)